



# Filtering of pulsed lidars data using spatial information and a clustering algorithm

Leonardo Alcayaga[1]

[1]DTU Wind Energy, Frederiksborgvej 399, 4000 Roskilde, Denmark

**Correspondence:** Leonardo Alcayaga (lalc@dtu.dk)

**Abstract.** Wind lidars present advantages over meteorological masts, including simultaneous multi-point observations, flexibility in measuring geometry, and reduced installation cost; but wind lidars come with the 'cost' of increased complexity in terms of data quality and analysis. Carrier-to-noise ratio (CNR) has been the metric most commonly-used to recover reliable observations from lidar measurements, but with severely reduced data recovery. In this work we apply a clustering technique to identify unreliable measurements from pulsed lidars scanning a horizontal plane, taking advantage of all data available from the lidars—not only CNR, but also line-of-sight wind speed ($V_{LOS}$), spatial position, and $V_{LOS}$ smoothness. The performance of this data filtering technique is evaluated in terms of data recovery and data quality, against both a median-like filter and a pure CNR-threshold filter. The results show that the clustering filter is capable of recovering more reliable data in noisy regions of the scans, increasing the data recovery up to 38% and reducing by at least two thirds the acceptance of unreliable measurements, relative to the commonly used CNR-threshold. Along with this, the need for user intervention in the setup of data filtering is reduced considerably, which is a step towards a more automated and robust filter.

## 1 Introduction

Long range scanning wind lidars are useful tools, and its adoption has grown rapidly in recent years in wind energy applications (Vasiljevic et al., 2016). Their capability to measure the evolution in time and space of atmospheric boundary layer wind fields in large spatial domains (which can reach a size comparable to a wind farm), their increasing accuracy and low cost of installation are important advantages over meteorological masts. One disadvantage of these devices with respect to traditional wind speed measurement techniques is the influence that atmospheric conditions and instrument noise have on the data quality. For long-range scanning lidars, this becomes an important issue due to the lack of references to identify reliable observations, since noise can contaminate large portions of the scanning domain. The most commonly used criteria to retrieve reliable observations is a threshold on values of the Carrier-to-noise ratio, CNR, which, depending on the site conditions, experimental setup and instrument manufacturer, can take values between $[-29$ dB, $-20$ dB$]$ and $[-8$ dB, 0 dB$]$ as lower and upper bounds, respectively (Gryning et al., 2016; Gryning and Floors, 2019). This criteria results in large amounts of data rejected unnecessarily in regions far from the instrument, due to the nature of CNR which decreases rapidly with distance. To cope with this issue Meyer Forsting and Troldborg (2016) and Vasiljević et al. (2017) have proposed filters based on the smoothness and continuity of the wind field. Such filters work by detecting discrete or anomalous steps in $V_{LOS}$ which present a difference





between of line-of-sight wind speed, $V_{LOS}$, and its local (moving) median above a certain threshold, predefined by the user. Beck and Kühn (2017) first and Karagali et al. (2018) in an adapted version, follow a different approach (here called KDE filter, from Kernel Density Estimate) based on the statistical self-similarity of the data, which, in simple terms, assumes that reliable observations are alike and will be located close together in the observational space. The probability density distribution
of observations (estimated via KDE) in a dynamically normalized $V_{LOS} - $ CNR space shows that measurements likely to be valid are located in a high density region. Observations sparsely distributed beyond a boundary defined by a threshold in the acceptance ratio, or the ratio between the probability density of any observation and the maximum probability density over the whole set of measurements, are finally identified as noise. Both approaches need the definition of one or more thresholds and a window size, either in time for the KDE filter, or in space for the wind field smoothness approach. These parameters are
dependent on different characteristics of the data, like the lidar scanning pattern for instance.

Both approaches miss important and complementary information, either neglecting the quality of acquired data (quantified in terms of CNR) or the spatial distribution and smoothness of the wind field. Moreover, in both apporaches the position of observations is not taken into account, information that can shed light on areas permanently showing anomalous values of $V_{LOS}$ or CNR, like e.g. hard targets. Complementing all these features within the smoothness approach is difficult, since CNR
is not a smooth field like $V_{LOS}$ and including them in the KDE filter increase the computational time substantially, since the basis to define high density regions and an acceptance ratio is a kernel density estimate in a higher dimensional space (3 or more features including spatial position and smoothness), which is computationally intensive due to the estimation of a bandwidth parameter and the definition of a high density region (hyper-volume) with a good resolution.

Data self similarity – over any scale in the case of fractals or a range of them in real situations (Mandelbrot, 1983) –
is closely related to clustering techniques (Backer, 1995), which are capable of classify large data sets with many different features at a relatively low computational cost. For instance, the KDE filter shares some characteristics with the popular $k$-$means$ clustering algorithm (first presented by MacQueen (1967)), since they define one (or several for $k-means$) specific group of data belonging to an unique category (or cluster) which size and location on the observational space will depend on data density or, more specifically, on a kernel density estimation. The main difference between these two algorithms is the way
they treat sparse data points that fall in low density regions. In $k-means$, sparse points are assigned to the cluster with the nearest center, no matter if they are outliers or present unlikely values from a physical point of view. The KDE filter solves this problem introducing an acceptance ratio, which corresponds to a threshold on the probability density of data points that are to be accepted as cluster members. This threshold must be defined a priori when used on unfiltered data. Additionally, $k-means$ needs to define the number of clusters present in the data beforehand, unlike the KDE filter, which defines always an unique
cluster of valid data points, centered at the origin of the scaled and normalized observational space.

The `DBSCAN` clustering technique (Ester et al., 1996; Pedregosa et al., 2011) presents several advantages over $k$-$means$ in detecting clusters in a higher dimensional space: 1) introduces the notion of noise/sparsely distributed observations and 2) it does not need prior knowledge of the number of clusters in the data and it is capable of identify clusters of arbitrary shape. To the best of our knowledge, this is the first time that this type of clustering algorithm is applied to identify not reliable





observations from pulsed lidars. This approach, which can be understood as a natural extension of the KDE filter, is compared to the smoothness based filter on two types of data: synthetic wind fields data as a controlled test case, and real data.

This paper is organized as follows: Section 2 describes the real data used to test the different filtering approaches, and Section 3 presents the synthetic data used during a controlled test as well as the methodology to obtain it. Section 4 then gives a description of the different filters applied in this study to both data sets, to continue with the definition of the performance tests

in Section 5. In Section  6 the performace tests are presented along with a discussion on their validity and quality. Section 7 discuses the quality of the methodology behind the tests and the advantages and disadvantages of the proposed approach. Section 8 presents the conclusions of this study.

## 2  Real data: Østerild Balconies experiment

The filtering techniques presented here were tested on lidar measurements made at the Østerild Test Centre located in north-
ern Jutland, Denmark, see Figure 1. The aim of this experiment was to characterize the horizontal flow field above a flat, heterogeneous forested landscape at two heights relevant for wind energy applications. Known as the Østerild Balconies experiment (Mann et al., 2017; Karagali et al., 2018; Simon and Vasiljevic, 2018), the wind speed data covers a large horizontal area (around 50 km$^2$), with the possibility of characterizing flow patterns in a wide range of scales, both in time and space. However, these advantages come with increased complexity on data reliability. A larger measurement area is affected by local terrain and
atmospheric conditions, like clouds or large hard targets. Moreover, at this scale lidars reach their measuring limitations, since the back-scattering from aerosols decrease rapidly with distance (Cariou, 2015).

The Balconies experiment consists of two measuring phases (see Table 1) with two long-range WindScanners performing Plan Position Indicator (PPI) scanning patterns, aligned in the North-South axis and installed at 50 m a.g.l. during phase 1 and 200 m a.g.l. in phase 2. WindScanners  (Vasiljevic et al., 2016) consist of two or more spatially separated lidars which
are synchronized to perform coherent scanning patterns, allowing the retrieval of two or three dimensional velocity vectors at diffeent points in space. These experiments were conducted between April and August of 2016 (Simon and Vasiljevic, 2018). In each phase, the northern and southern lidars scanned in the West and East direction relative to the corresponding meteorological masts, where they were installed. The data used in this study originated from both phases of the experiment, with PPIs pointing to the west. For more details about the experiment, lidars and terrain characteristics see (Karagali et al., 2018; Vasiljevic et al.,
2016; Simon and Vasiljevic, 2018).

## 3  Synthetic data

Assessing and comparing the performance of filters is challenging with no reference available to verify that rejected or accepted observations are truly outliers or simply wrong observations. This is especially difficult for long-range scanning lidars, since their measurements cover large areas and, due to spatial variability, a valid reference would need several secondary
anemometers scattered over the scanning area. Testing filters on a controlled and synthetic data set, contaminated with a well





**Table 1.** Characteristics of the Balconies experiment, from Karagali et al. (2018). The scans are not instantaneous neither totally synchronous, with a horizontal sweep speed of 2°/s in the azimuth direction in a range of 90°, with a total time of 45 s per scan.

| Phase | Measurement start | Measurement end |
|---|---|---|
| 50 m a.g.l. (1) | 2016-04-12 12:45:41 | 2016-06-17 12:48:01 |
| 200 m a.g.l. (2) | 2016-06-29 13:35:56 | 2016-08-12 09:09:55 |
| **Scanner** | **Location coordinates, [m]** | **Scanning pattern, west** |
| Southern lidar | 492768.8 (East) 6322832.3 (North) | 344°-256°, 2° steps |
| Northern lidar | 492768.7 (East) 6327082.4 (North) | 196°-284°, 2° steps |

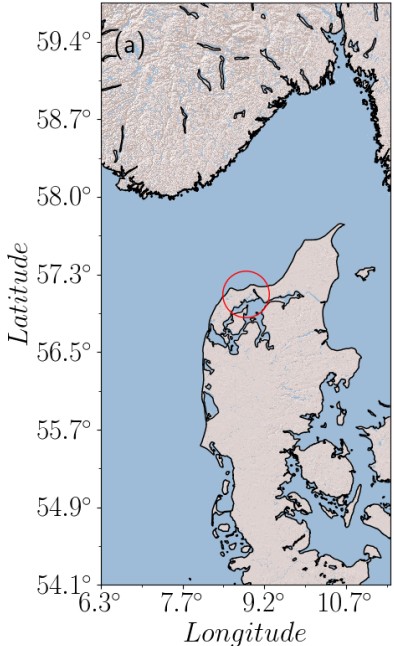

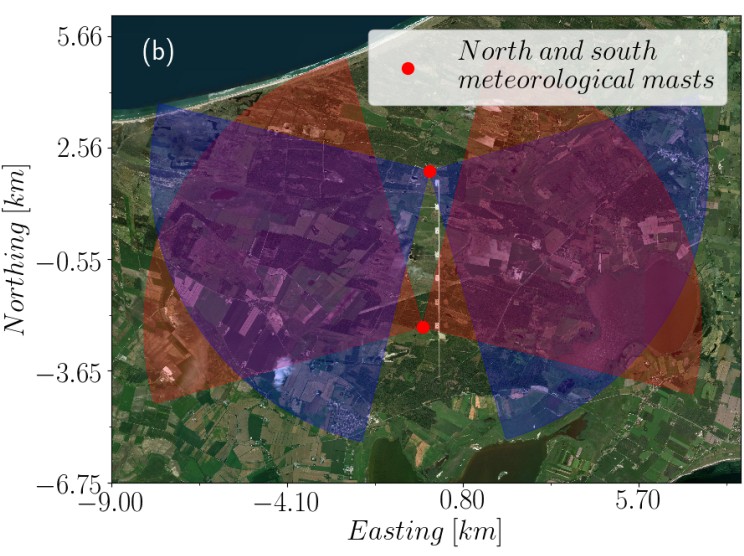

**Figure 1.** (a) Location of the Østerild Turbine Test Center, place of the Balconies experiments, northern Jutland, Denmark (copyright 2009 Esri). (b) Detail of the test center site, with the location of the meteorological masts where north (blue) and south (red) WindScanners were installed. During the measurement campaign the PPI scans pointed both west in some periods and both east in other (copyright 2017 DigitalGlobe, Inc.).

defined noise, presents an option to deal with this problem. In this study, the filters presented in Section 4 are tested on individual scans sampled from synthetic wind fields generated using the Mann turbulence spectral tensor model (Mann, 1994), and contaminated with procedural noise (Perlin, 2001).



**Table 2.** Synthetic wind field characteristics and parameters.

| Parameter | Values |
|---|---|
| $L$, m | 62, 125, 250, 500, 750, 1000 |
| $\alpha\epsilon^{2/3}$, m$^{4/3}$s$^{-2}$ | 0.025, 0.05, 0.075 |
| $\Gamma$ | 0, 1, 2, 2.5, 3.5 |
| Number of seeds used | 10 |
| Mean wind speed, $U$ m/s | 15 |
| Mean wind speed direction range, degrees | 90 to 270 |
| Total number of scans generated | 4305 |

## 3.1 Synthetic wind fields generation

Synthetic PPI scans are sampled by a numerical lidar from synthetic wind fields generated via the Mann-model (Mann, 1998) in a horizontal, two-dimensional square domain of 2048 x 2048 grid points, with dimensions 9200 m x 7000 m. The generated turbulence fields are the result of input parameters of the of turbulence spectral tensor model: length-scale, $L$, turbulence energy dissipation $\alpha\epsilon^{2/3}$, and anisotropy, $\Gamma$. The fields generated correspond to wind speed fluctuations, which are subsequently added later to a mean flow field to generate the resulting synthetic wind speed fields. An initial random seed generate different wind

field realizations with the same turbulence statistics. For details on wind field generation using the the Mann-model, refer to Mann (1998). Table 2 shows the range of values used for the generation of two-dimensional wind fields. Large values of $\alpha\epsilon^{2/3}$ or smaller scale turbulence for instance, make that sudden spatial changes in wind speed are more likely, which increase the false identification of outliers. Mean wind direction, turbulence anisotropy and length scale will also affect the sampling due to the lidars sampling characteristics.

## 3.2 Sampling with a numerical lidar

The numerical model of a long-range pulsed lidar attemps to mimic what the real instrument does obtaining, for instance, the data described in Section 2. Even though its implementation is very crude and simplified, it allows to generate sampled scans with reasonable spatial smoothness and complexity, but it is, however, not able to reproduce either noise nor CNR values. Since the interest of this study is to test filters over a large number of scans, its simplicity allows a quick sampling over high

resolution synthetic wind fields, thus reducing computational time. The sequence followed by the model in sampling from the high resolution wind fields is specified below (see also Figure 2 for more details):

1.  A coarse two dimensional mesh in polar coordinates $(r, \theta)$ is generated, with radial (range gates) and azimuth ranges and resolution described in Table 3.

2.  A set of non overlapping nested meshes are then constructed locally, centered in each point in an upper level, coarser

mesh. Each nested mesh has 21 x 51 grid points in the radial and azimuth directions, respectively. The resolution in the





radial direction of the nested mesh guarantees at least 1 synthetic observation within grid elements located in the two beam range gates closest to the lidar (the closest range gate in this case is 150m from the lidar), since, even though the resolution of the wind field Cartesian grid is high, the size of its elements is comparable with the nested ones close to the numerical lidar. The number of elements in the azimuth direction allows a low grid aspect ratio at the end of the beam, which is 7km from the lidar.

3. The streamwise and lateral wind speed components from the synthetic wind field, $U$ and $V$, are linearly interpolated into the nested fine mesh and the radial component ($V_{LOS}$) calculated using (1), with $\theta$ being the azimuth angle of the local beam.

$$V_{LOS} = \cos(\theta)U + \sin(\theta)V \tag{1}$$

4. The numerical lidar mimics the volume averaging in a real lidar by averaging in the radial direction each of the 51 beam segments in the local grid, using $V_{LOS}$ values weighted with $w$ defined by

$$w = \frac{1}{2\Delta p}\left\{\text{Erf}\left[\frac{(r-F)+\Delta p/2}{r_p}\right] - \text{Erf}\left[\frac{(r-F)-\Delta p/2}{r_p}\right]\right\} \tag{2}$$

as in Banakh and Smalikho (1997) and Smalikho and Banakh (2013). Here $F$ is the distance from each point in the beam to the corresponding range gate, $\Delta l$ is the lidar beam's full width at half maximum (provided by the lidar's manufacturer), $\Delta p$ is the range gate length (cf. Table 3),

$$\text{Erf}(x) = \frac{2}{\sqrt{\pi}}\int_0^x \exp(-t^2)dt \tag{3}$$

denotes the error function, and

$$r_p = \frac{\Delta l}{2\sqrt{\ln(2)}} \tag{4}$$

is the beam width contribution to the volume averaging. This kernel is not truncated, having influence of back-scatter from more distant positions on the local estimation, thus smoothing the $V_{LOS}$ final field. Even though $w$ decays rapidly with distance for the numerical lidar setup used here, there is still some influence from points located around distant range gates.

The result of this beam averaging is one radial velocity, $V_{LOS}$, per range gate, 51 values in total per nested grid, along one arc in the azimuth direction.

5. Pulsed lidars accumulate information of the back-scattered signal spectra as they sweep an azimuth sector (2 degrees in our case) before estimation of the spectral peak and $V_{LOS}$. This continuous sweep is modeled here by the 51 discrete values of $V_{LOS}$ along the azimuth arc and their non-weighted average. This step generate radial wind speeds in each point of the initial coarse polar grid, which constitutes the synthetic scan.





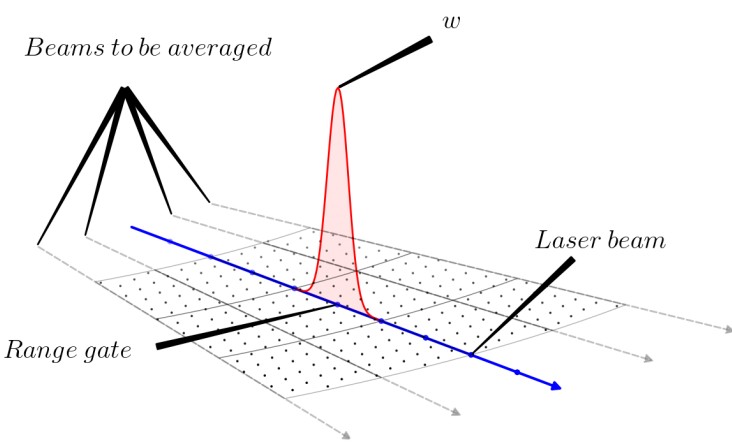

**Figure 2.** The numerical lidar mimics the accumulation of information in the spectra of the back-scattered signal (within each $2°$ step) as the average of 51 discrete beams. The radial wind speed in each of the 51 local range gates is the result of a weighted contribution of 21 points along each local beam segment, which spans from the previous to the nest range gate. The final $V_{LOS}$ will be the average of these 51 values. Here the weight, $w$, for beam averaging, originates from equation (2).

The local fine polar mesh and averaging in the azimuth direction here play the role of the rapid sweeping of the laser

beam, taking into account that the measured $V_{LOS}$ will be the result of the accumulation of information on the back-scattering spectra within the $2°$ azimuth steps, as well as the radial/line averaging around each range gate. This model is, of course, very simplistic and does not contain information about back-scattering spectra nor Doppler effect. Moreover, it does not mimic the non-instantaneous nature of the real scans, but rather assumes that all beams cross the measuring domain simultaneously, recording radiual wind speeds at the same time. Nevertheless, it will generate scans with radial speeds projected from a realistic

wind field and can be contaminated with noise (presented in Section 3.3) under controlled conditions, which is the final goal of this test.

### 3.3   Synthetic noise generation

The most simple noise that can be used to contaminate synthetic scans is sparse, uniformly distributed outliers, taking extreme values from the tails of non reliable observations (see Figure 10), and always within the detectable range of $\pm$ 35 m/s, char-

acteristic of the pulsed lidar described in Section 2. This noise, also known as salt and pepper noise, is easily detected and eliminated by median-like filters, when extreme discrete steps affect the smoothness of an image (Huang et al., 1979; Burger and Burge, 2008). Nevertheless, what one can see from noise in real scans are regions of anomalously high and/or low $V_{LOS}$ values (see Figure 4 (a)), and, depending on their relative size to the moving window of a median-like filter (more details en



**Table 3.** The characteristics of the numerical and real (Karagali et al., 2018; Vasiljevic et al., 2016) long-range lidars used for the controlled test of the filters.

|  | Numerical | Real |
| --- | --- | --- |
| Azimuth range | $256°$ - $344°$ | $256°$ - $344°$ |
| Azimuth step | $2°$ | $2°$ |
| Beam length | 7000 [m] | 7000 [m] |
| Range gate length | 35 [m] | 35 [m] |
| Time per scan | Instantaneous | 45 [s] |
| Coarse polar grid size (radial-azimuth) | 45 x 198 | - |
| Nested (local) polar grid size (radial-azimuth) | 21 x 51 | - |

Section 4), they could pass through the filter undetected and unchanged. A more realistic noise (and also a more difficult to
detect, goal of the test), is the procedural noise generated via the simplex noise algorithm, introduced by (Perlin, 2001) origi-
nally to recreate more natural synthetic textures on surfaces for computer graphics applications. This type of noise is an option
to generate more natural noisy regions with smoother transitions between large $V_{LOS}$ outliers. The basic principle behind this
algorithm can be roughly summarized as follows:

1. A relatively coarse two dimensional grid of pseudo random unit gradients is generated. The pseudo-randomness is
established as follows: a list of permutations (indexes permuted) with the same number of elements as the grid will
    sample gradients from a list of limited length (16 unit gradient elements in our case).

2. A set of points $p$ is arbitrarily distributed within the domain defined by the grid.

3. To estimate the noise level of each point $p$, the contribution from each of the nearest pseudo-random gradients is cal-
    culated as the dot product between the corresponding gradient and the distance vector, **d**, from the gradient position to
$p$. Finally, each contribution is added directly after weighting by the inverse of the magnitude of **d**, and scaled to obtain
    values within the range [-1,1].

4. As one could suspect, each $p$ matches one of the positions to contaminate in the synthetic scans.

When clouds, rain or atmospheric conditions affecting the concentration of aerosols in the air enter the measuring domain,
what we see are regions with anomalous line-of-sight wind speeds and low CNR values. One of the advantages of procedural
noise is the generation of areas with noise, with the same size of the scan domain for instance, either in polar or Cartesian
coordinates, via an array of points $p$ distributed over the synthetic measuring domain. In this test, the distribution of points $p$
over the scanning area aims to follow the decay in the back-scatter intensity with distance (See Figure 5). We define three bands
per scan, centered at 50% , 70% and 90% of the total beam length, spanning over the entire azimuth range and with a width of
30% of the beam length in the radial direction. Within each of these bands a set of uniformly distributed positions are selected.





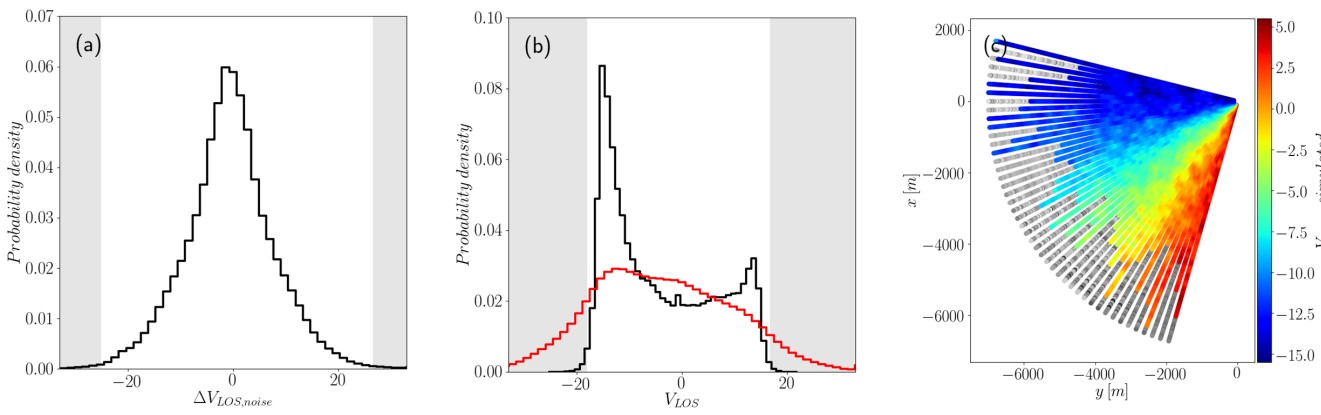

**Figure 3.** Contamination with procedural noise on synthetic scans.(a) Distribution of $\Delta V_{LOS,noise}$, the noise added to the original scan. Maximum values are within the observable range between $-35$ and $35$ [m/s]. (b) Distribution of non-contaminated $V_{LOS}$ (black) and contaminated, $V_{LOS} + \Delta V_{LOS,noise}$ (red), for all range gates and cases. The distribution of $V_{LOS}$ show peaks around $\pm15$ m/s, which correspond to the main wind directions simulated. (c) Individual scan showing a realization of the spatial distribution of radial wind speed, $V_{LOS}$, with increasing fraction of added noise $\Delta V_{LOS,noise}$ with distance.

The fraction of beams contaminated at each band, as we depart from the lidar, are 30%, 60% and 90%, respectively, since it is assumed that after a position of a particular beam is sampled, the remaining points in the radial direction are all contaminated. The increasing fraction of contaminated points tries to be similar to what is observed in real data, since the quality of the signal decrease as we depart form the lidar. The noise amplitude is finally scaled by 35 [m/s]; i.e. the limit of the observable range of $V_{LOS}$ for the instruments described in 2.

Figure 3 (a) and (b) show the distribution of the noise generated by the simplex algorithm after scaling, and its effects on the distribution of $V_{LOS}$, respectively. The distribution of contaminated observations presents heavier tails, as expected, but also a high probability for $V_{LOS}$ values within a range far from the shadowed area of outliers. Therefore, contaminated points are more difficult to spot by filters that do not consider information about spatial smoothness. Figure 3 (c) shows one contaminated synthetic scan, where the noise fraction increases with distance. The noisy areas show relatively smooth transitions in the

azimuth direction due to the grid of gradients is generated in polar coordinates, which makes the noise identification more difficult for median-like filters, which are good at detecting local discrete steps in $V_{LOS}$, but have difficulties to do it when the size of its moving window is comparable to the size of the contaminated area.



## 4    Filtering techniques tested on real and synthetic data

### 4.1    CNR threshold

Along with the line-of-sight wind speed, lidars give information about the intensity of the back-scattered signal via CNR, which can take values in a range from 15 to -50 dB, depending on the manufacturer of the instrument. Here, the low CNR values correspond to very poor signal back-scattering, due to atmospheric conditions (i.e. low concentrations of aerosols) and high ratios usually occur when the lidar laser beam hits a hard target (Cariou, 2015). Good and reliable measurements will lay in between these too scenarios, and thresholds values have traditionally been used to filter out non reliable observations (Gryning

et al., 2016). Observations with CNR values lying beyond upper and lower thresholds (which will depend on the lidar itself, commonly -8 and -24 dB, respectively) are rejected. One problem that arise with the use of this filtering criteria for long range lidars is that the CNR value worsen rapidly with distance, and observations which might be valid at spatial points that are distant from the instrument are rejected using this criteria. Figure 4 (b) shows a scatter plot of radial wind speed, $V_{LOS}$, against CNR, from 30 consecutive PPI scans from the Balconies experiment, with its typical comb-like shape. Below the lower threshold (in

this case -27 dB) along with extreme values of $V_{LOS}$ we also find line-of-sight speeds that are not far from the reliable range above the threshold. The effect of filtering out only observations below the threshold or accepting the the ones with values within the range of $V_{LOS}$ above the threshold can be seen in Figure 4 (c) and (d). Low CNR values for reasonable line-of-sight speeds can be understood when the range distance is included in the picture, as in Figure 5: the decay in the back-scattering intensity with distance makes CNR values worse, but, apparently, the lidar is still able to retrieve some valid observations. The

selection of an appropriate threshold for CNR is not clear, but a lower bound of -29 dB is recommended by Gryning and Floors (2019) before a wind resource assessment, based on lidar measurements, starts to be dependent on the CNR-threshold.

### 4.2    Median-like filter

The main output of scanning lidars is, in the present measuring campaign, a two-dimensional field of line-of-sight wind speeds. Interpreting this field as an image, a median filter arise as a viable option for detecting erroneous measurements, since it is

well known that this type of non-linear filter is suited to detect and filter noise that present distributions with large tails Huang et al. (1979). Here we use an adaptation of the traditional median filter used in the image processing community: values are not replaced by the local moving median but labeled as reliable or non reliable according to whether their values are either under or above a threshold for $\Delta V_{LOS, threshold}$, the absolute difference between the value and its moving median. Another difference is the two-dimensional nature of the original median filter, which estimate the local median within a two dimensional moving

window. Here, the median-like filter does the same but in two one-dimensional instances, the first in the radial direction, $r$, and finally in the azimuth direction, $\theta$, considering the polar coordinates of the scan. This simplification reduces the computation time, which is the main advantage of this filter. The input parameters of this filter will be the size (or number of elements) of the moving window in the radial direction, $n_r$, the size of the window in the azimuth direction, $n_\phi$, and a threshold for the difference between the local radial wind speed value and its corresponding moving median, $\Delta V_{LOS, threshold}$. For fixed values

of $\Delta V_{LOS, threshold}$, $n_r$ and $n_\phi$, the spatial structure of wind speed fluctuations will have an effect on the recovery rate of this

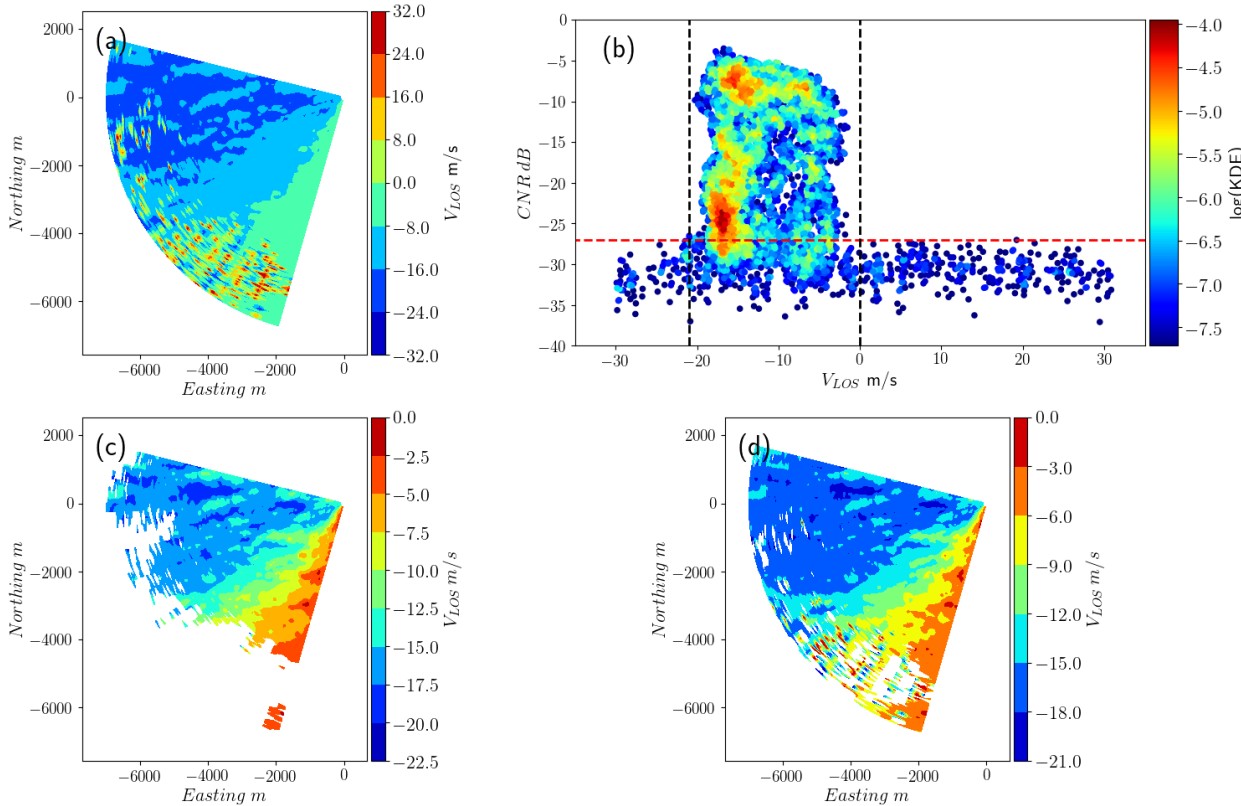

**Figure 4.** (a) Scan from data measured in the Østerild Balconies Experiment with noisy observations in the far region. (b) Distribution of CNR and $V_{LOS}$ for the same scan, including the probability density estimated via KDE. Observations with high CNR values show a limited range of $V_{LOS}$ (black dashed line). A CNR threshold (dashed red line) of -27 places a portion of reliable observations that belong to a high probability density region in the rejection area. (c) Since points with low CNR are located in the far region of the scan, a large portion of the measured area is lost after filtering. (d) Values out of the range defined by the dashed black lines in (d) are rejected instead. The measured area gained still shows some noisy observations

filter. Section 6.1 explores this relationship by means of a sensitivity analysis on the performance of this filter when applied over wind fields with different characteristics.

### 4.3 Filtering using a clustering algorithm

Assuming no abrupt spatial changes in the different features measured, radial wind speed for instance, observations non affected by poor back-scattering or noise generated in the lidar itself, will fall in limited regions of the observational space, unlike contaminated observations, that will be scattered in wider region. This is noticeable in Figure 5, distinguishing two main groups: one presenting limited radial wind speeds and CNR values (between -20 and 0 m/s and 0 and -35 dB, respectively) and another group, less dense and broader, with bad observations. Also noticeable is an overlapping region that contains






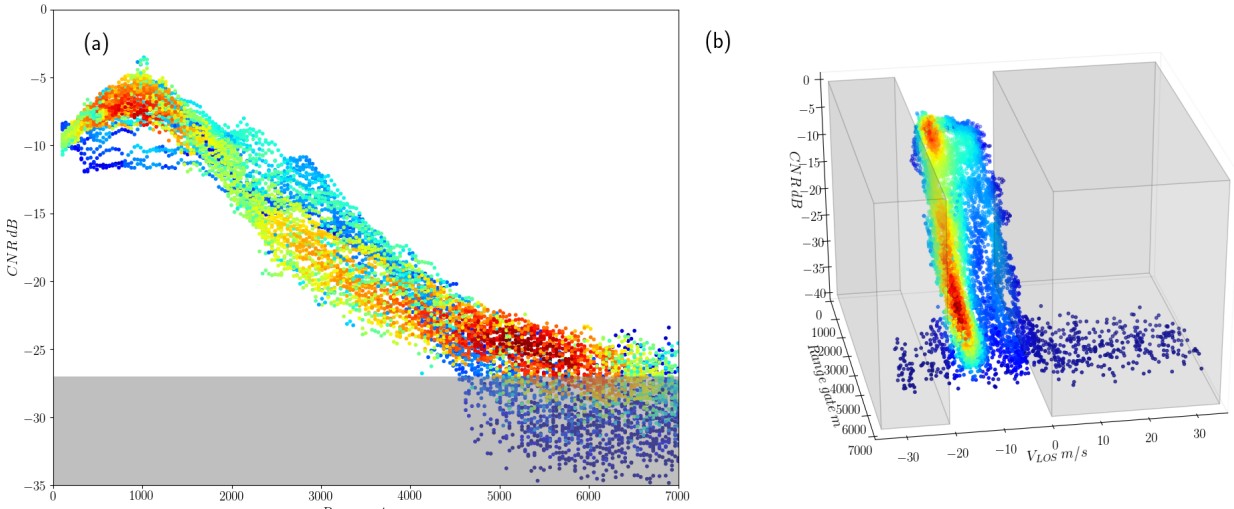

**Figure 5.** (a) When range gate distance is included in the picture, the shape of the distribution of reliable observations gets more complicated, and a decreasing CNR with distance is now obvious. Observations with low CNR values but high probability density can be found at the more distant region of the scan. (b) Threshold in the actual values of $V_{LOS}$ (shadowed volumes) recovers distant, good measurements, but still includes non reliable observations with very low probability density or KDE values

observations from both groups. When only $V_{LOS}$ and CNR are taken into account (Figure 4 (b)), the overlapping region is more diffuse than when including the range gate distance in the picture (Figure 5). The two groups are more distinct in the latter representation because their separation increases along with dimensionality as we consider more features of the data. These considerations inspire us to think the identification of valid observations as a process, which classifies groups or *clusters* of good/bad data, using all the information available: temporal and spatial information, signal quality via CNR, $V_{LOS}$ values and its smoothness.

There exist many clustering algorithms with different characteristics and performance (Rui Xu and Wunsch, 2005; Xu and Tian, 2015), each of them suitable to the specific characteristics of the data being analyzed. Partitions algorithms, like *k-means* (MacQueen, 1967) or *k-medoids* (Park and Jun, 2009) have been popular in the data mining community due to its low computational complexity. In these algorithms data points are separated in $k$ different groups or clusters, with each observation belonging to the cluster with the closest mean/medoid, estimated iteratively by minimization of the within-cluster variance. Even though these algorithms introduce the notion of density in the data distribution, they have two drawbacks: both need prior knowledge of the number of clusters present in the data and, if this first problem is sorted, the algorithm assigns all data points to specific clusters, either good measurements or outliers, which is not desirable for our purpose. The Density Based Spatial Clustering for Applications with Noise algorithm, or DBSCAN (Ester et al., 1996), on the other hand, is a clustering technique specially designed to deal with large-scale data with spatial distribution. When compared to other algorithms, DBSCAN presents several advantages when applied as a filter to measurements from lidars: 1) it can manage large amounts of data with spatial





distribution (its time complexity is $O(n \log n)$, with $n$ the amount of $m$-dimensional points in the dataset), 2) it does not need prior knowledge of the number of clusters in the data, 3) it is a non-supervised algorithm (meaning that it does not require any fitting with previous training data), 4) it identifies clusters with arbitrary shapes and 5) it does introduce the notion of a noise set, which are data points that do not belong to any cluster.

`DBSCAN` identifies clusters and noise based on two parameters, the neighbourhood size, $\varepsilon$, and the minimum number of nearest neighbours, NN. The parameter $\varepsilon$ is the euclidean distance to the limits of a neighborhood in which NN (or more) nearest neighbors of each point may be located. Intuitively, these parameters will define the minimum density that a data partition needs to have to be identified as a cluster. In formal terms, clusters are defined as a collection of *density connected points*, the most general category out of four:

– Core point: A point $q$ is a core point if within its $\varepsilon$-neighborhood we can find NN or more points apart from $q$.

 – Direct density reachable point: A point $p$ is directly density reachable from a core point $q$ if $p$ is in the $\varepsilon$-neighborhood of $q$.

 – Density reachable point: A point $p$ is density reachable from $r$ if there exist a set of core points $q$ directly connected between them and to $r$ and $p$.

– Density connected points: Points $p$ and $r$ are density connected if they are density reachable for at least one common core point $q$.

The euclidean distance here is $m$-dimensional, because the data is characterized by $m$ features. Since the values taken by each feature can be very different ($V_{LOS}$ is in a range from -35 to 35 m/s, and range gates can be between 105 and 7000 m apart from the lidar, for instance), each feature of the data needs to be centered and scaled properly before the application of
`DBSCAN` to obtain a meaningful distance between points. By using the mean value for centering and the standard deviation for scaling, this step is very sensitive to the presence of outliers. Therefore, in our case, the mean is replaced by the median and the scaling is done using the inter-quartile range instead, which is the range between the 25th and 75th percentiles.

Figure 6 shows schematically how this algorithm works. After centering and scaling, all $m$-dimensional distances between points are calculated, and `DBSCAN` starts traveling across all data points identifying them with the categories already presented:
core points, or points with NN neighbours within the $\varepsilon$-neighbourhood, border points, or density reachable points that do not meet the NN requirement but still have at least one core point within the $\varepsilon$-neighbourhood, and points classified as noise, or points that are not density connected. Finally, the algorithm define clusters as the individual groups of data with only density connected points and an unique group with points classified as noise, not belonging to any cluster.

The parameters $\varepsilon$ and NN have a significant influence on the number and characteristics of the clusters detected by `DBSCAN`.
Large values of $\varepsilon$ together with small NN will define very sparse clusters, including noise as valid cluster members. On the contrary, the requirement of small, densely populated neighborhoods will reject many points that otherwise are valid cluster members. As we can see, both parameters are closely related, and therefore one can fix one of them to vary the one remaining. Following the guidelines of the author of `DBSCAN` in Ester et al. (1996) NN is fixed to 5 neighbours in this case, no matter the



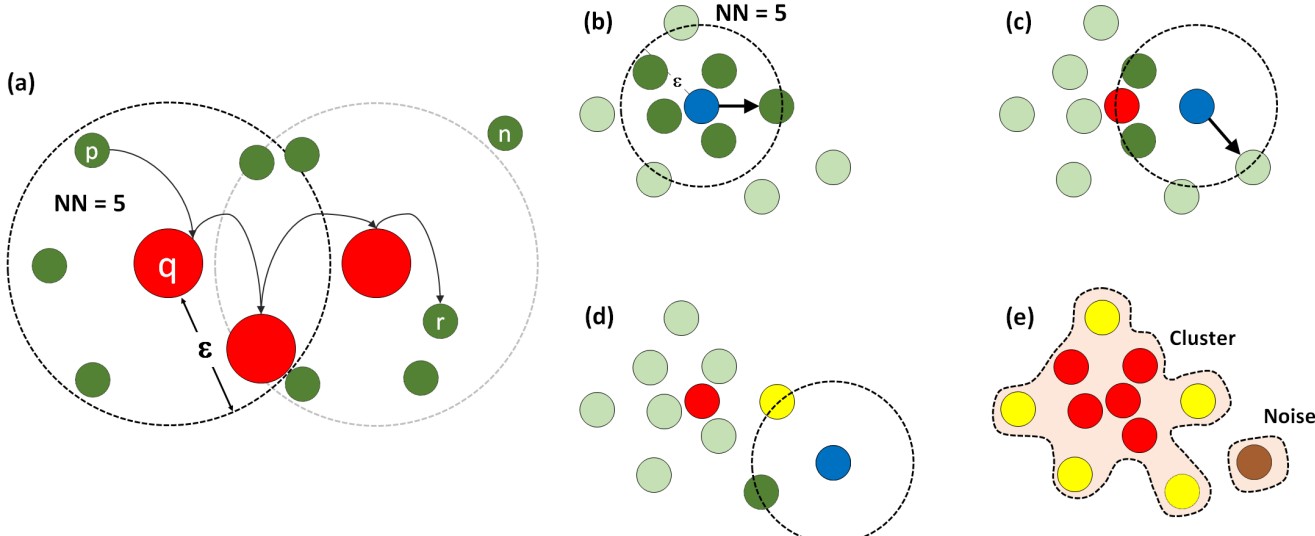

**Figure 6.** (a) `DBSCAN` algorithm definitions: direct density reachable point *p* (reachable by the core point *q*) and density reachable and density connected points *p* and *r*. Here point *n* does not belong to any of these categories, but noise. The `DBSCAN` algorithm working: (b) The current point being evaluated have the minimum number of nearest neighbours required, NN, within a neighborhood of size $\varepsilon$, classified as a *core point* (red) (c) The next point have less than NN neighbours, but one of them is a core point and becomes a *border point* (yellow) (d) A point with neither NN neighbours, nor core points within $\varepsilon$, classified as *noise* (brown) (e) The final cluster and noise. The former is a collection of density connected points.

type of data or its distribution, leaving only $\varepsilon$ to be estimated according to the structure of the data. One way to describe this data structure is via the *k-distance* function, $d_k(n)$. This function maps and sorts in ascending order the distance from each data point to its $k$-th nearest neighbour. Figure 8 (a) gives an idea of how this function looks when applied on real data. When the distance to the fifth nearest neighbour ($k = NN = 5$) is considered we can distinguish four turning points, or **knees**: the first, positive, at very small distances, the second, highlighted, represents the limit of reliable observations and non-reliable, and two more, both in the region where $k$-th distance grows faster within the non structured group of data, identified as noise. These knees in *k-dist* represents sudden changes in the data density, and they separate clusters from each other and from mere noise. One way to determine the position of these knees is to locate the corresponding peaks in the curvature of the $d_k(n)$ function, $\kappa(n)$. Even though $d_k(n)$ is discrete, expression (5) defines $\kappa(n)$ from its continuous analog, in which the primes correspond to the derivatives with respect to the "continuous" data point number $n$. The continuous representation of the discrete $k - dist(n)$ function is made by spline-fitting on a reduced set of uniformly distributed points over the original data set. By doing this, one can avoid the selection of local spikes not representing the general structure of the data.

$$\kappa = (d_k(n))'' / (1 + (d_k(n))'^2)^{3/2} \tag{5}$$



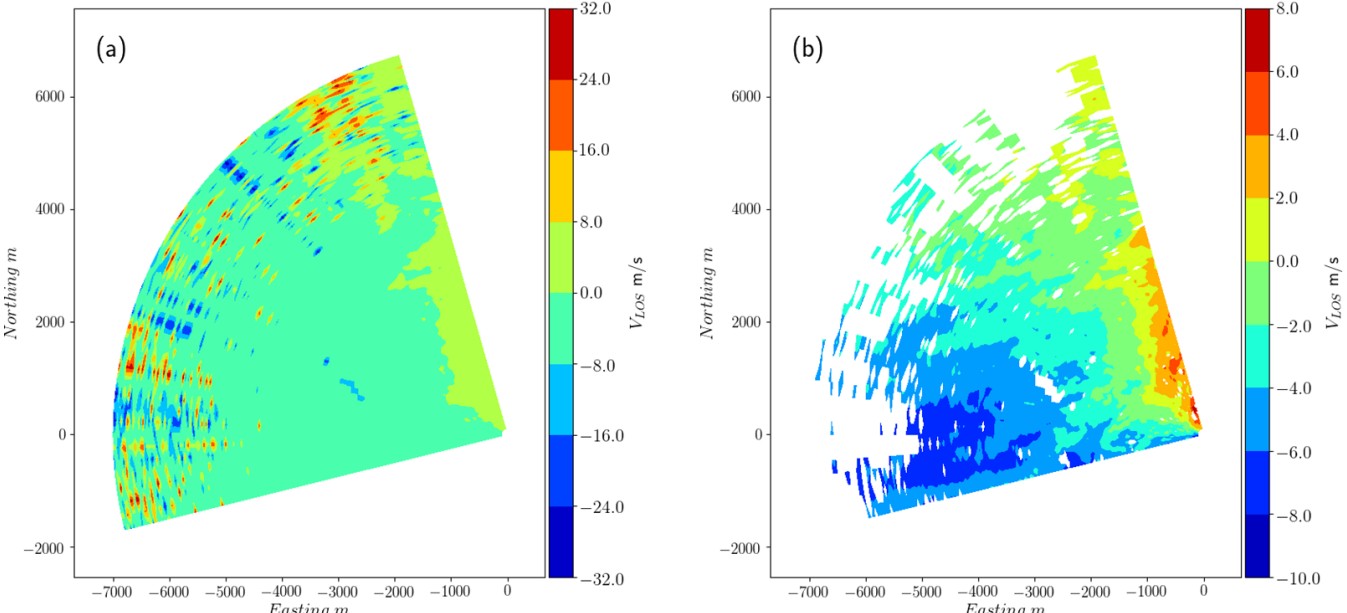

**Figure 7.** (a) Scan from the Balconies experiment (phase 1) with a 48% of data points in the range of reliable observations with CNR $\in$ [-24, -8] dB (b) The same scan after filtering with `DBSCAN` using $V_{LOS}$, range gate, azimuth angle, CNR and $\Delta V_{LOS}$ as features.

For scans representing good measurements the relative distance between these knees is well marked and easy to identify. This is not the case when scans are very nosy, like the one in Figure 9, and the positions of knees become closer, with a large number of observations showing a fast growing $k$-th distance. The selection of a $\varepsilon$ value that defines a reliable cluster of good observations is in this case is difficult but can be eased when the fraction of points with a reliable CNR value is also taken into account. In this case the neighborhood size is not selected by $\varepsilon_{knee}$, which is the $\varepsilon$-distance corresponding to the first noticeable knee from left to right, but by $\varepsilon_{CNR}$, defined by (6). Here $f_{CNR}$ corresponds to the fraction of very reliable observations in the data set (or measurements showing CNR values within the range [-24, -8]) over the total number of points. The constants $c_1$ and $c_2$ are defined by upper and lower bounds of $\varepsilon$, defined as the values that $k - dist(n)$ takes at the first and last knee from left to right in Figure 9 (a), respectively.

$$\varepsilon_{CNR} = c_1 f_{CNR} + c_2 \qquad (6)$$

Regarding the features considered to characterize each data point, depending on whether we filter synthetic or real data, these will be radial or line-of-sight wind speed, $V_{LOS}$, azimuth and radial positions and $\Delta V_{LOS}$. The latter corresponds to the median difference in radial wind speed of an specific radial and azimuth position with its direct neighbours. This feature is included to consider the smooth spatial fluctuations of the radial wind speed, same assumption used by the median-like filter. The real data will include $CNR$ as an additional feature. The controlled performance test on synthetic data does not include this feature, because the numerical lidar described in Section 3 just estimates $V_{LOS}$.





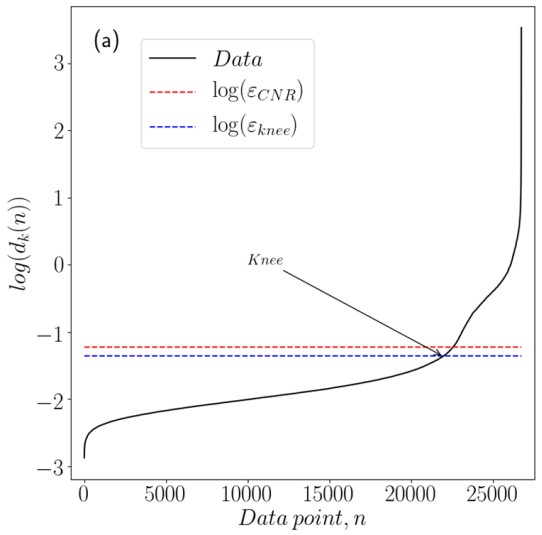

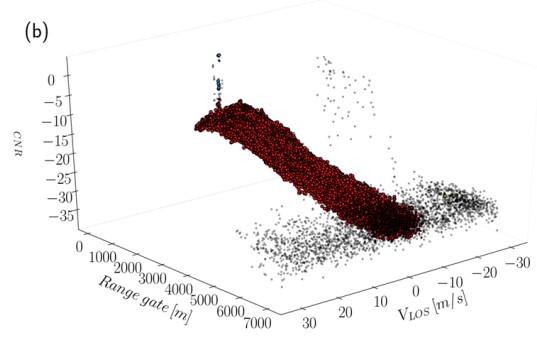

**Figure 8.** Data structure of the scan shown in Figure 7. (a) Logarithm of sorted distances to the $5$-$th$ nearest neighbour for each point in a data set. The total number of observations corresponds to three consecutive scans, or 26730 points. The sorted $5$-$th$ distances show three knees separating three types of structures: reliable observations with distances below $\varepsilon_{knee}$, an overlapping region where the distance between points grows faster and pure noise or non structured data. (b) Tri-dimensional representation of the data (range gate, CNR and $V_{LOS}$), where two coherent structures of data points, or clusters, are identified. The large structure has distances between observations below $\varepsilon_{knee}$ and corresponds to reliable observations. The vertical line showing high CNR values are observations of bad $V_{LOS}$ measurements in a group of 7 turbines around 2000 m from the lidar (see Figure 16).

## 5 Performance metrics

### 5.1 Synthetic data

The advantage of testing the filters on controlled cases is the prior knowledge of the position and magnitude of noise. This allows us to define three metrics to assess how the median-like and clustering filters perform, namely, 1) fraction of noise detected $\eta_{noise}$, 2) fraction of good observations recovered $\eta_{recov}$ and 3) a total performance metric $\eta_{tot}$, which takes into account the relative importance of contaminated observations, $N_{cont}$, and non-contaminated observations, $N_{non-cont}$, using the noise fraction $f_{noise}$ as weight of the two metrics $\eta_{noise}$ and $\eta_{recov}$, which are complementary as a large noise detection

fraction will have associated a lower recovery fraction of good measurements. In expressions (7) to (9) this metrics are defined in formal terms, where $N_{noise}$ is the number of observations identified as noise by the filter, $N$ the total number of observations, and $N_{pos}$ and $N_{neg}$ the total number of false positives and false negatives, respectively.

$$\eta_{noise} = \frac{N_{noise} - N_{pos}}{N_{cont}} \tag{7}$$





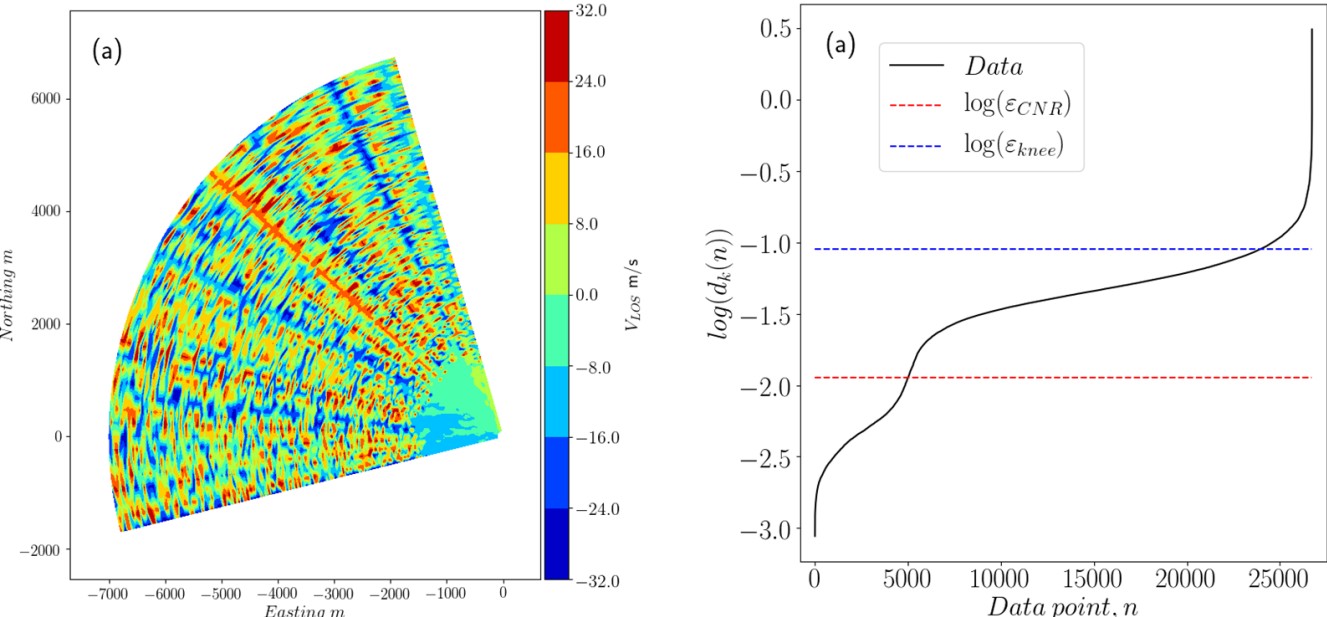

**Figure 9.** (a) Scan from phase 1 of the experiment with a 13% of data points in the range of reliable observations with CNR $\in$ [-24, -8] dB (b) Data structure of the very noisy data. Here $\varepsilon_{knee}$ over estimates the neighbourhood distance for a coherent cluster, and the inclusion of noisy measurements is avoided via $\varepsilon_{CNR}$

$$\eta_{recov} = \frac{N - (N_{noise} + N_{neg})}{N_{non-cont}} \tag{8}$$

$$\eta_{tot} = f_{noise}\eta_{noise} + (1 - f_{noise})\eta_{recov} \tag{9}$$

## 5.2 Real data

Unlike the controlled test, in real measurements we do not have any explicit reference to validate the results from the two filters. The quality of the data retrieved after their application is assessed by comparing the distribution of radial wind speeds for very reliable observations (with CNR values within the range between -24 to -8 dB) with the distribution of filtered observations that fall out of this range. Observations out of the reliable range population usually show a probability density function (or pdf) with heavier tails, like the pdfs in Figure 10. Here we understand a heavy tailed pdf as a distribution that slowly goes to zero and show higher probability density for values beyond the 3-$\sigma$ limit (or 3 standard deviation limit), when compared to the normal distribution, evidence of a higher probability of occurrence of outliers or extreme values. The recovering rate of observations beyond the [0.003, 0.997] quantile range of the reliable $V_{LOS}$ (shaded area in Figure 10) could shed information about the quality of the data retrieved by the filter.





Other metric is the similarity between pdf of reliable and non reliable data, after filtering. The distance between both probability density functions can be compared with similarity metrics like the Kolmogorov-Smirnov test (Kolmogorov, 1933) or

Kullback–Leibler (KL) divergence (Kullback and Leibler, 1951). The former test measures the statistical similarity between two random variables, $X_1$ and $X_2$, by estimating the statistical distance, $D$ (or K-S statistic), between their cumulative distribution functions, $F_1(x)$ and $F_2(x)$, as the supreme of their difference,

$$D_K = \sup_x \|F_1(x) - F_2(x)\| \tag{10}$$

The null hypothesis here is that two realizations from the same distribution, if the K-S statistic is such that its two tailed

$p$-value is above a certain level $\alpha$. Due to the amount of data analyzed here is huge—we analyzed over 20000 scans for the two phases of the Østerild campaign, each with 8910 data points, over almost 10 days—this similarity test is very precise, but also very strict rejecting the null hypothesis for small deviations between $F_1(x)$ and $F_2(x)$. Nevertheless, the K-S statistics can be used to compare which probability distribution is closer to the one representing the reliable observations: the non-reliable observations after filtering with 1) the median filter or 2) the clustering filter approach.

The KL divergence is a measure of similarity, or overlapping of two distributions $P_1$ and $P_2$ , with realizations $X_1$ and $X_2$, respectively. It is used in different applications to shed light on the loss of information when $X_1$ is represented by $P_2$ or vice-versa and is defined by the expression (11).

$$D_{KL} = \sum_x P_1(x) \log\left(\frac{P_2(x)}{P_1(x)}\right) \tag{11}$$

Both metrics will be used to estimate how the distribution of non reliable observations of $V_{LOS}$ is modified after filtering,

and if the new distribution is similar (or close, in a statistical distance sense) to the probability density of reliable observations of the radial wind speed, shown in Figure 10 for phases 1 and 2 of the measurement campaign, respectively.

Both performance metrics, the recovery rate of abnormal measurements in the tails of the pdf of reliable observations and its statistical distance to the pdf of filtered non reliable observations, will be assessed for the median-like filter, the clustering filter and also for data filtered with a CNR threshold of -29 dB, following (Gryning and Floors, 2019).

# 6 Results

## 6.1 Synthetic data

As described in section 4.2, the median filter needs three parameters as input, $n_r$, $n_\phi$ and $\Delta V_{LOS, threshold}$, which will have a large impact on its outcome. For a given window size, both in radial and azimuth directions, a small value of $\Delta V_{LOS, threshold}$ will affect the capacity of the filter to retrieve observations that are valid but reflect large local wind speed fluctuations, affecting

the data recovering rate. A larger value of this parameter in the other hand, will result in many noisy observations that are not the result of turbulent fluctuations being accepted as valid. The synthetic data set with 4305 scans and different turbulence characteristics allow us to test different combinations in these three parameters and find the optimal filter, which can be used



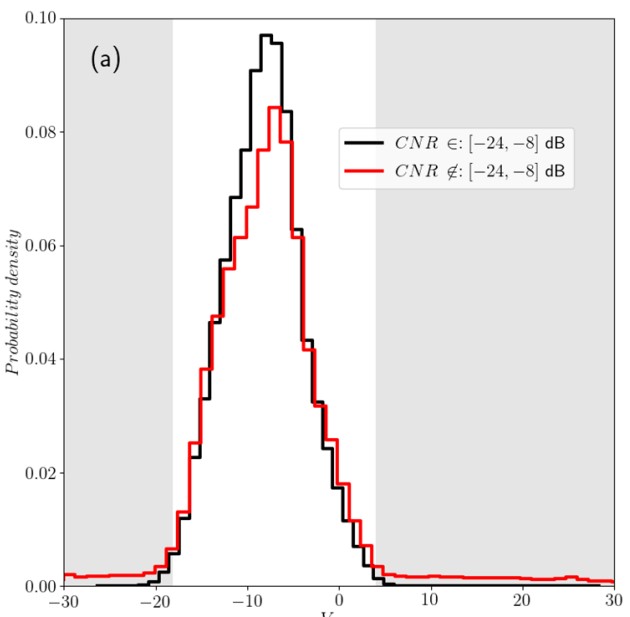
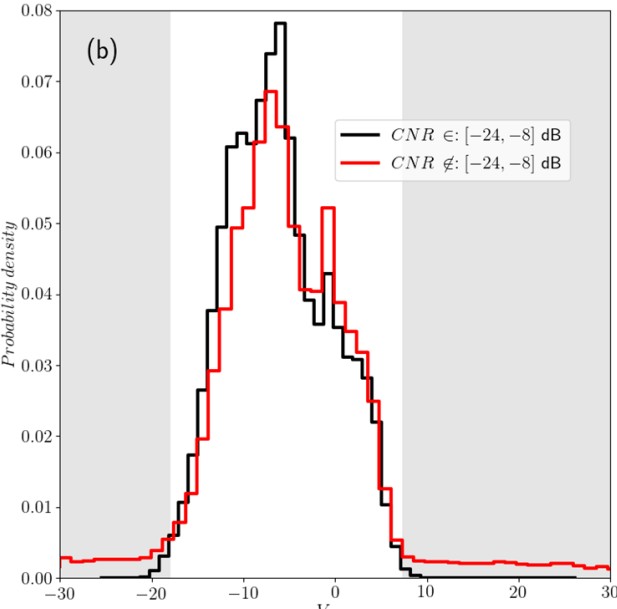

**Figure 10.** Probability density function of reliable observations of $V_{LOS}$ (black solid line) and non reliable observations (red, solid line) for (a) Phase 1 of Balconies experiment with scans performed at 50 m a.g.l. and (b) Phase 2 of the same campaign, with scans performed at 200 m a.g.l.

to compare against the results of the clustering algorithm. The set of parameters used for this purpose are within the range of odd values from 3 to 13 elements for $n_r$, $n_\phi$ and in the interval 1 to 6 m/s for $\Delta V_{LOS, threshold}$. Figure 11 shows contours

that present the most optimal value for $\eta_{tot}$ among all possible values of $\Delta V_{LOS, threshold}$ and $n_\phi$, for $n_r$ = 5, the optimal window size in the radial direction. Large $\Delta V_{LOS, threshold}$ results in large $\eta_{recov}$ but poor results for $\eta_{noise}$ and the opposite for values of the threshold, as expected. The metric $\eta_{tot}$ then becomes relevant then to determine the optimal combination of parameters. From the contours it is possible to see that the performance in terms of the $\eta_{tot}$ metric is less sensitive to $n_\phi$ than $\Delta V_{LOS, threshold}$. Even though the results here show average metrics for all the scans simulated, the optimal value of

$\Delta V_{LOS, threshold}$ increases with the turbulence energy and length scale parameters, which is problematic, because it requires previous knowledge of turbulence characteristics that usually are not available before reconstruction, and more important, data filtering. In order to compare the performance of the median-like filter to the clustering filter, the optimal set $n_r$ = 5, $n_\phi$ = 3 and $\Delta V_{LOS, threshold}$ = 2.33 m/s will be used.

The clustering filter uses $V_{LOS}$, the azimuth and radial positions, $r$ and $\phi$, and $\Delta V_{LOS}$. Due to the nature and simplicity of

the numerical lidar implemented, it is not possible to generate synthetic values of CNR, and this feature can not be included to characterize the data points. The clustering filter is implemented to be a non-supervised classifier, and does not need more input parameters than the different features and the number of scans put together as a batch before filtering. The latter is set to three in this case, to speed up calculations and avoid creating clusters from noisy regions. From this point of view, this





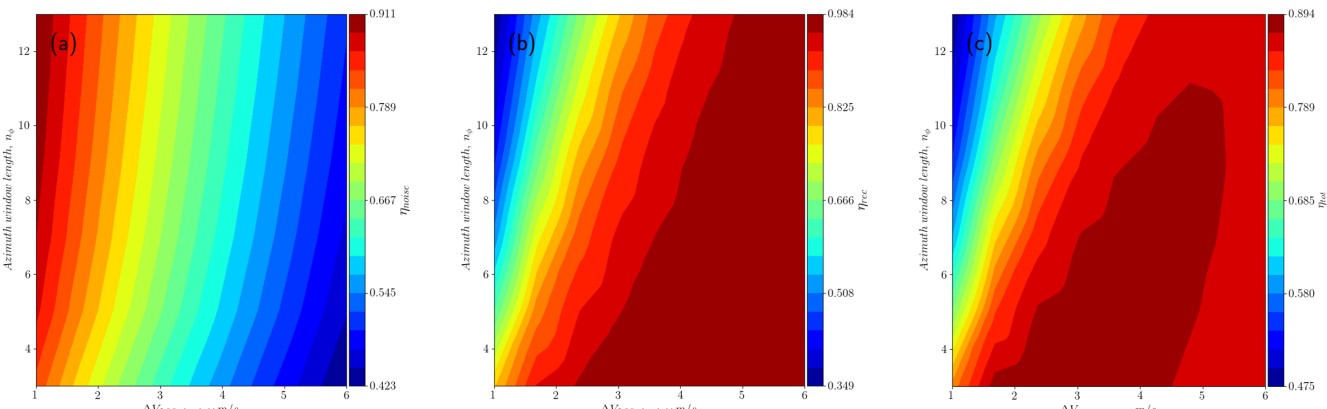

**Figure 11.** Contours of performance metrics for $n_r = 5$ over the $\Delta V_{LOS,\,threshold}$-$n_\phi$ space. Each point in in the contour plot corresponds to the mean value of (a) $\eta_{noise}$, (b) $\eta_{rec}$ and (c) $\eta_{tot}$ among all the 4305 synthetic scans filtered. The optimal value corresponds to $n_r = 5$, $n_\phi = 3$ and $\Delta V_{LOS,\,threshold} = 2.33$ m/s

filter is also dynamic as that of Beck and Kühn (2017) when applied to a real data set, since it will consider the data structure

within a period limited to 135 seconds (3 scans of 45 seconds in our case), and characteristics of temporal evolution of the data is indirectly taken into accoount. For the synthetic data used in this test, it is useful to have more than one scan filtered per iteration and have enough density of data points in both, noisy and reliable areas of the observational space. We speculate that scans that are correlated in time will enhance the self-similarity of the data, thus improving the performance of the filter. Turbulence structures with length scales in a range between the range gate size and the scanning area size will evolve at a

slower rate than the time elapsed between consecutive scans. These structures will be then present at positions that are not very distant from its previous location in the earlier scan. The consequence of this is more dense clusters for good observations than when we consider independent scans realizations.

Figure 12 shows the resulting metrics of the two filters applied on the synthetic data set. Looking at $\eta_{tot}$, both filters show similar mean values and spread, with the clustering filter performing slightly better. The difference becomes noticeable when

we see $\eta_{noise}$, which for the clustering filter show a mean value of 0.95, far larger than the 0.67 of the median-like filter. The latter result could be problematic if the median-like filter is used, since noise contaminating the filtered scan will result in non realistic wind fields after reconstruction.

Both filters perform well when evaluated in terms of $\eta_{rec}$, with the median-like filter showing a higher mean fraction of good observations retrieved, 0.96, compared with the 0.89 of the clustering filter. This result is expected, since the median-

like filter is more permissive regarding fluctuations that can seem locally anomalous for the clustering filter. It is not clear that the recovering rate of the clustering filter will be benefited by including more features from the data set. The euclidean distance used by the clustering algorithm to identify nearest neighbours increases (to a certain level) as we add more features or dimensions the data description. As a consequence, adding more information to the data set, like CNR for instance, will result in less dense clusters and noise, making it easier to identify noise and improving the noise detection ratio, but on the other hand



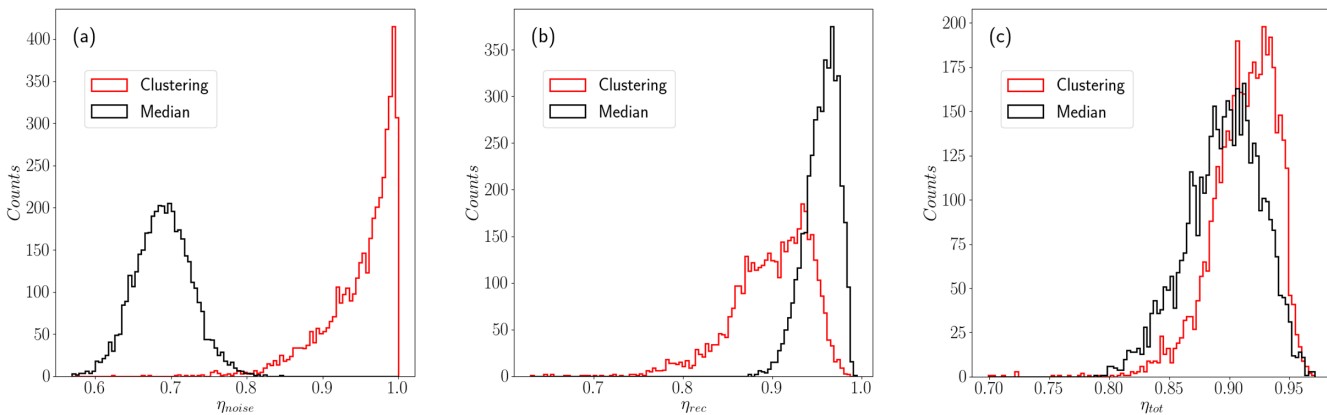

**Figure 12.** Histograms of the three performance indexes for the total number of synthetic scans (a) Both filters show similar spread but the clustering filter rejects a rather higher fraction of noise. (b) The higher recovery rate of the median-like filter, and its narrower distribution is superior than the clustering algorithm, the cost is acceptance of more contaminated observations (c) Both filters have similar mean values for $\eta_{tot}$ around 0.9

making the identification of good measurements located near the border of a cluster more difficult, especially if the CNR value of a good measurements is too low/high, due to poor scattering/closeness to hard targets. This might be solved including more scans per filtering iteration, but this can not be tested with the synthetic data, given the simplified numerical lidar implemented here.

## 6.2   Real data

The data set from the Balconies experiment presents advantages for the clustering filter, since the CNR value can be included as a feature in describing the data. Nevertheless, as mentioned already in section 2, we do not count on any reference to asses the performance of the filter apart from the radial wind speeds distribution of very reliable observations with CNR values within the range between -24 dB and -8 dB. As mentioned earlier, valid observations in this range might present a similar distribution. Figure 10 shows this distribution before filtering, shadowing the area of values of $V_{LOS}$ that fall in the region

beyond a 99.7% of the total probability or $3\sigma$ limit, usually classified as outliers. Figures 13 and 14 show the recovery fraction for CNR, median-like and clustering filters when applied on data in the reliable and non reliable CNR ranges for phases 1 and 2 of the Østerild experiment. Unlike the clustering filter, the CNR threshold and median-like filters show non negligible recovery rates beyond the $3$-$\sigma$ limit, particularly significant in the former. This result is very much in line with the $\eta_{noise}$ metric from the synthetic data. Within the $3$-$\sigma$ range, the CNR and median-like filters perform slightly better than the clustering filter in

terms of recovery fraction, in agreement with the results of $\eta_{rec}$ in section 6.1. Even though this might compensate the fact that CNR threshold and median-like filters fail to filter out the major part of outliers, increasing the availability of measurements, this difference does not make the pdf of the filtered data more similar to the pdf of reliable data, as Table 4 shows via the metrics $D_K$ and $D_{KL}$. According to this metric, the pdf of the data after the application of the clustering approach looks more



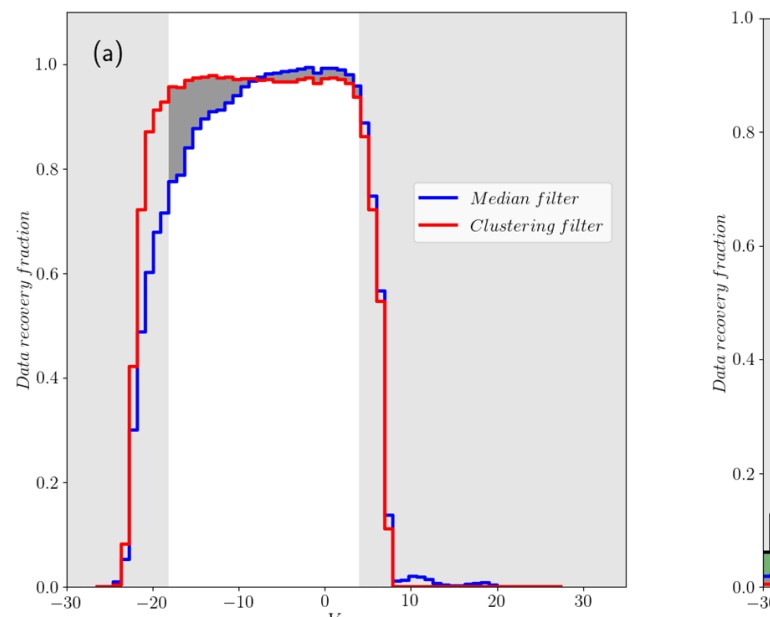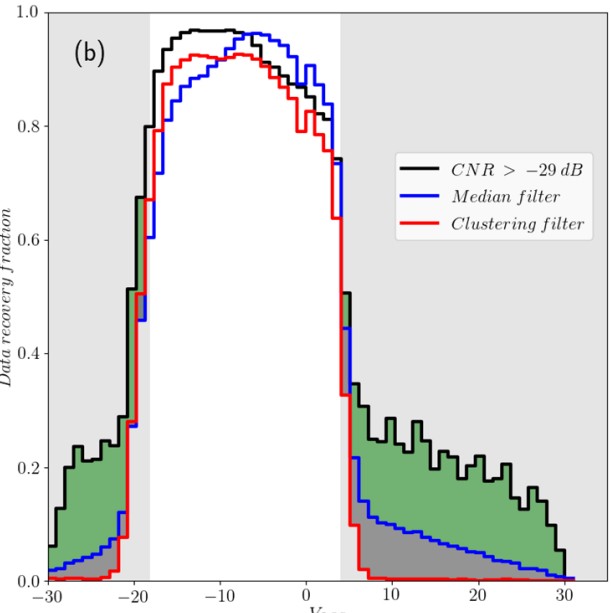

**Figure 13.** Distribution of recovery fraction per wind speed bin for phase 1 of the experiment of (a) reliable observations ($-24 <$ CNR $< -8$) and (b) non reliable data (CNR $< -24$ or CNR $> -8$) for the three types of filter. The shadowed area in both graphs corresponds to the region where observations exceed the 99.7% of probability (or 3-$\sigma$ limit) in the pdf of reliable observations. The darker shadowed areas highlights the additional fraction of extreme values non-filtered by the median-like and CNR filters, when the former uses the optimal input set $n_r = 5$, $n_\phi = 3$ and $\Delta V_{LOS, threshold}$) = 2.33 m/s.

statistically similar to reliable observations. This table also show $D_K$ and $D_{KL}$ of the non reliable data before filtering, which

in all cases is improved, except for $D_K$ for median and CNR threshold filters during phase 2.

Figures 15, 16 and 17 show the performance of the three different filters in different regions of the scan, from respectively phase 1 and 2 of the experiment. When the spatial distribution of the recovery fraction is analyzed, we can see that the lowest values shown by the clustering filter are mostly located in the far region of the scan which, in general, presents low CNR values. The spatial recovery rate during phase 1 also show that the median-like and clustering filters are able to identify hard

targets, which are also a source of bad observations. For scans recorded at 50 m above ground level in phase 1, back-scatter is affected by a group of seven turbines located approximately in the middle of the scanning area, with one turbine touching the end of the southern beams of the scan and a meteorological mast located very close to the lidar. Figure 16 shows a detail of the recovery rate associated with the flow in the vicinity of the turbines group, in which we can see that the clustering filter is able to identify better the turbine locations, recovering more data in the sorroundings when compared to the median-like filter.

Table 5 shows a summary of the additional data available when the CNR = -29 dB threshold, the median-like and the clustering filters are applied instead of the more conservative and restrictive CNR = -24 dB threshold filter. Additionally, this table shows the fraction of observations exceeding the 3-$\sigma$ limit that are recovered by the three filters. Even though the





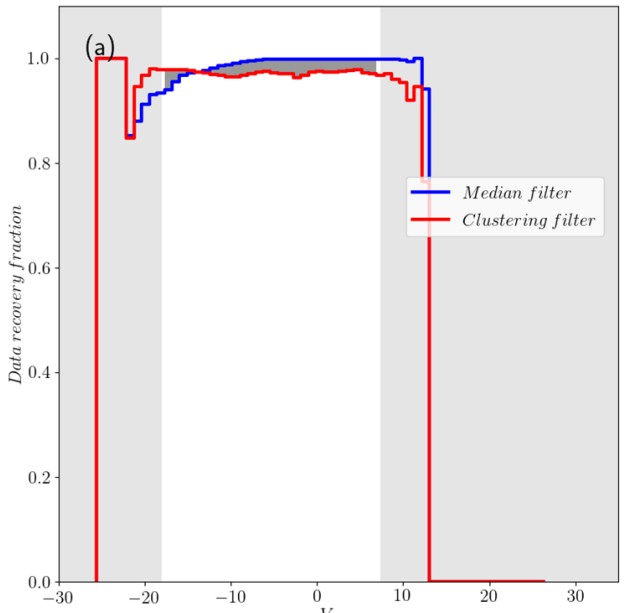 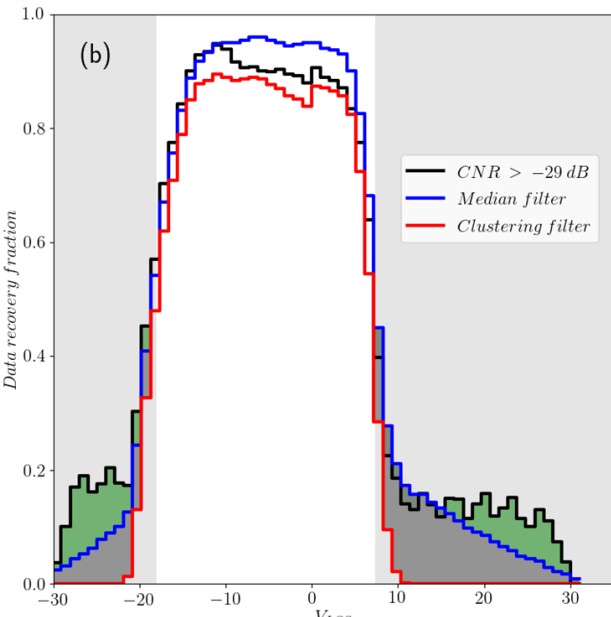

**Figure 14.** Distribution of recovery fraction per wind speed bin for phase 1 of the experiment of (a) reliable observations and (b) non reliable data for CNR, median and clustering filter. The shadowed area in both graphs corresponds to the region where observations exceed the 3-$\sigma$ limit in the pdf of reliable observations. Again, darker shadowed areas highlights the additional fraction of extreme values non-filtered by the median-like and CNR filters, when the former uses the optimal input set $n_r = 5$, $n_\phi = 3$ and $\Delta V_{LOS,\,threshold}$ = 2.33 m/s.

clustering filter shows a slightly lower fraction of additional data available when compared to the other filters, most of it comes from values within the 3-$\sigma$ region. Moreover the quality of the data recovered by the clustering approach seems to be higher

when all these results are tested with the performance metrics defined in Section 5.

## 7   Discussion

### 7.1   Performance assessment on synthetic data

The metrics introduced in section 6.1 attempt to evaluate two different capabilities of the filters: the quality of the data recovered and the amount of good quality data recovered. In general these two metrics are in conflict—specially for the median-like

filter—every time a high rate of noise is removed also good measurements will be removed. The metric $\eta_{tot}$ attempts to quantify their relative importance regarding the noise fraction, which in this study is distributed in a relatively wide range, but on average represents 20% of the total number of measurements per scan. The impact of the noise fraction distribution on the performance of the filters was not explored, and variations on its dispersion and mean value might be necessary. Regarding the synthetic scans, they do not allow the identification of outliers in the time domain because they are time independent. A time





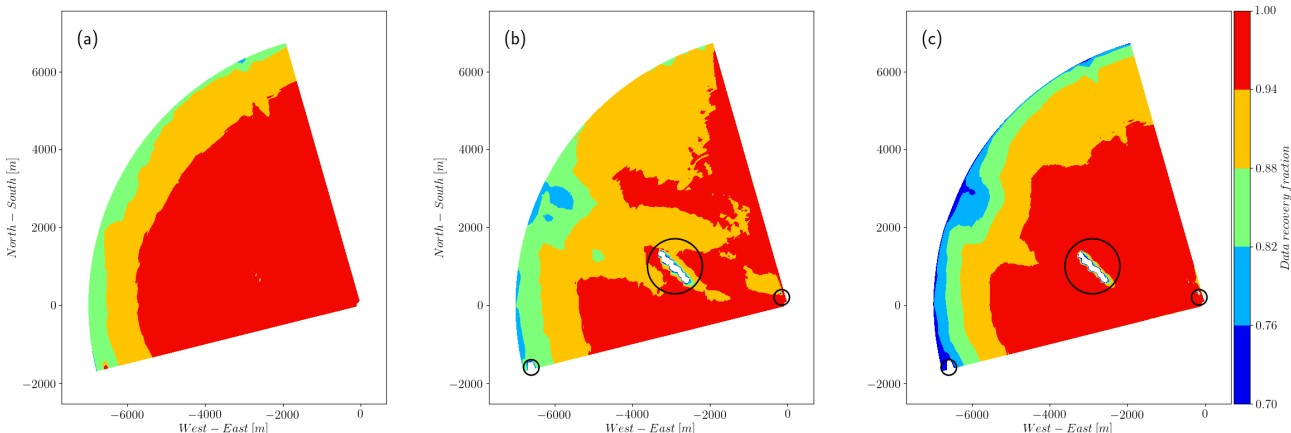

**Figure 15.** Total recovery fraction for phase 1 of the experiment. The noisy and far region of the scans show a high recovery, above 80%, for (a) the CNR > -29 dB threshold filter and (b) the median-like filter and below 75% for (c) the clustering filter. Highlighted, it is possible to see three groups of hard targets (turbines and one meteorological mast, close to the lidar), which are identified by the median and clustering filter with recovery rates below 20%.

**Table 4.** Results of pdf similarity test of reliable and non-reliable data after filtering. The CNR = -29 dB threshold is also included by Gryning and Floors (2019)

| Phase 1 | $D_K$ | $D_{KL}$ |
|---|---|---|
| Non-reliable data before filtering | 0.097 | 0.134 |
| CNR threshold > -29 dB | 0.045 | 0.109 |
| Median filter | 0.047 | 0.126 |
| Clustering filter | 0.037 | 0.105 |
| Phase 2 | | |
| Non-reliable data before filtering | 0.110 | 0.126 |
| CNR threshold > -29 dB | 0.114 | 0.052 |
| Median filter | 0.117 | 0.057 |
| Clustering filter | 0.103 | 0.045 |

evolving synthetic turbulence fields would be necessary to generate scans correlated in time and enhance the self similarity of the data. This might improve the performance of the clustering approach and allow the addition of a time dependence in the median-like filter, used already in Meyer Forsting and Troldborg (2016).

The synthetic wind fields used here do not consider the presence of hard targets. These anomalies in the wind field are observed by lidars as points with high CNR values and abnormal $V_{LOS}$. Assessing the performance of the filters in detecting





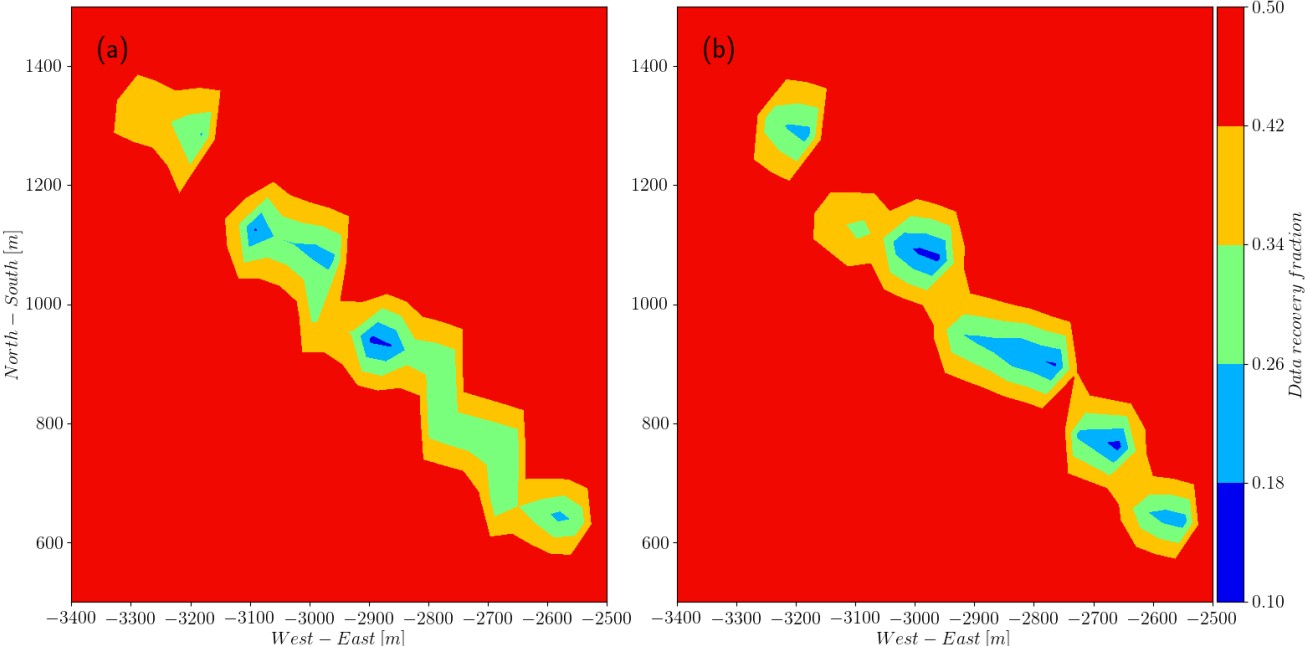

**Figure 16.** Detail of the recovery rate at the site of the turbines for (a) median filter and (b) clustering filter. The recovery is lower in the flow regime of the turbines cluster (there are 7 turbines in line) and higher in their surrounding for the clustering filter. Red denotes recovery rates of 0.5 or higher.

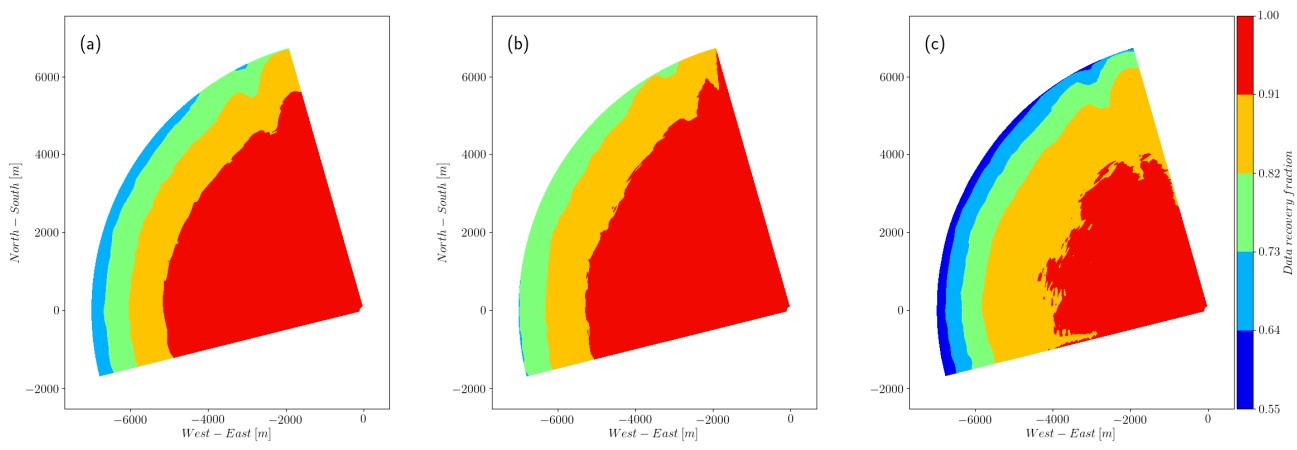

**Figure 17.** The total recovery fraction of observations for phase 2 of the experiment. The noisy and far region of the scans show a high recovery, above 70% for (a) and (b), the CNR > -29 dB threshold and median-like filters, respectively. The recovery decreases to 55% in the same region for the clustering filter, in line with the previous results, assuming that outliers (above the $3\sigma$ limit) and noise are more likely to be located here.





**Table 5.** Additional data recovered, relative to the amount of observations in the reliable range of CNR, and fraction of data recovered with values beyond the 3-$\sigma$ range.

| Phase 1, $3\sigma$ quantiles = [-18.16, 3.96] m/s | Fraction of data recovered beyond $3$-$\sigma$ | Additional data recovered |
|---|---|---|
| CNR threshold > -29 dB | 27.1% | 23.4% |
| Median filter | 14.0% | 23.1% |
| Clustering filter | 8.6 % | 22.1% |
| Phase 2, $3\sigma$ quantiles = [-18.08, 7.35] m/s | | |
| CNR threshold > -29 dB | 16.5% | 40.4% |
| Median filter | 12.6% | 42.4% |
| Clustering filter | 3.2% | 38.1% |

such anomalies needs a more realistic model of the pulsed lidar. This numerical lidar would allow the generation of information normally available in real lidar measurements, like CNR, and the spread in the power spectra of the heterodyne signal, $S_b$. This additional information will benefit the performance assessment of the clustering filter and the simulation of hard targets. A more realistic lidar model was already implemented by Brousmiche et al. (2007), which can be used to explore further these aspects of the filtering process.

**7.2 Performance assessment on real data**

The data set analyzed from the Balconies experiment corresponds to horizontal scans at 50 and 200 m above the ground level, limiting the analysis to one scanning pattern. Different scanning patterns, in vertical and horizontal planes, as well as wind fields over different topography would make this analysis more general, thus shedding light on the capabilities of the filters here presented. This is specially critical regarding the median-like filter, which might require again a sensitivity analysis to 465 select proper parameters that adapt to different scanning patterns and turbulence field characteristics. So far, $\Delta V_{LOS, threshold}$ showed a dependence on the $L$ and $\alpha\varepsilon^{2/3}$ parameters during the sensitivity analysis presented in Section 6.1. Larger fluctuations in the $V_{LOS}$ field, whether they come from larger turbulent structures or higher turbulence energy or both, will need a larger value of $\Delta V_{LOS, threshold}$ to avoid the rejection of good measurements. Range Height Indicator (RHI) scanning patterns can pose the challenge of strong vertical shear and small turbulent structures that will need to reduce the window size $n_r$ and $n_\phi$ 470 for the median-like filter, and the selection of a different set of features (or a new definition for $\Delta V_{LOS}$) for the clustering filter, in order to keep reliable observations from being filtered out.

Regarding feature selection, the clustering filter could consider the spectral spreading of the heterodyne signal, $S_b$ and time variation of $V_{LOS}$, in addition to features already used in this work to characterize and distinguish better cluster of good measurements. Nevertheless, due to the Euclidean distance definition, additional dimensions will make the data more sparse 475 in higher dimensions, making it necessary to use more data points per filtering step (here we used only 3 scans at a time) to





avoid the identification of good observations as spread, low density noise. It is because of this that the application of a feature selection method might be necessary (Chandrashekar and Sahin, 2014).

Finally, using the statistical distances $D_K$ and $D_{KL}$ as a metric for the filter performance might not be totally correct. At range gates far from the lidar, the distance between beams increases, as well as the area covered by the accumulation of spectral

information in azimuth direction. Averaging $V_{LOS}$ over larger areas as we move forward through each beam, might affect the statistics and the pdf of $V_{LOS}$ (specially its spread) in the outer region of the scan. The fact that this region is where we usually find the non reliable measurements group (the one having CNR values out of the range between -24 and -8 dB), may make the pdfs of reliable and non reliable observations somewhat different. This possible deviations need to be investigated further.

### 7.3   Advantages and limitations of proposed filters

From the results obtained in the analysis of real and synthetic data, the clustering filter show in general a better performance in noise removal and the recovering of good quality data from regions in the scan with poor CNR values. Moreover, this filter is based in a non-supervised clustering algorithm and requires little intervention from the user to obtain reliable results. This is a step forward to a more robust and automated processing of data from lidars, which ideally should be independent of the turbulence characteristics of the measured wind field or the scanning pattern used. The latter should be tested on a different

data set, as mentioned earlier.

The selection of features and the amount of scans put together per filtering step/iteration could also be automatized, using feature selection methods. Nevertheless, this would make the clustering filter more complex in its implementation and more computationally expensive, which is the main disadvantage of this approach compared to the median-like filter. Very efficient median filters can achieve a computational complexity up to $\mathcal{O}(n)$, with $n$ being the number of observations in the data set.

Depending on the data structure, DBSCAN shows a computational complexity from $\mathcal{O}(n\log(n))$ to $\mathcal{O}(n^2)$. If the distance between points is in general smaller than $\varepsilon$, the first limit can be achieved, but clusters with different densities makes the algorithm less efficient. In the data analyzed here, having clusters with different densities is not an issue. Nevertheless, for non homogeneous flows, scans might persistently show regions with $V_{LOS}$, CNR or other feature with noticeable different values, may need to revisit the clustering algorithm used and implement a $\varepsilon$-independent clustering approach, like OPTICS (Ankerst

et al., 1999) for instance.

### 8   Conclusions

The CNR threshold filtering has been the common approach to retrieve reliable observations form lidars measurements. In this work we compared this approach against two alternative techniques: a median-like filter, based on the assumption of smoothness of the wind field, hence, in the smoothness of the radial wind speed observed by a wind lidar, and a clustering filter,

based in the assumption of self-similarity of the observations captured by the wind lidar and the possibility of clustering them in groups of good data and noise. A controlled test was carried out on the last two approaches, using a simple numerical lidar that sampled scans from synthetic wind fields, later contaminated with procedural noise. The results indicate that the clustering



filter is capable of detecting more added noise than the median-like filter, at a good recovery rate of non contaminated data. When the three filters are tested on real data, the clustering approach shows a better performance on identifying abnormal observations, increasing the data availability between 22% and 38% and reducing the recovery of abnormal measurements between 70 and 80% when compared to a CNR threshold. Even though the median-like filter is computationally efficient, it needs an optimal definition of input parameters, which are dependent on the turbulence characteristics of the wind field. The clustering filter is more robust in this sense, because it is capable of automatically adapt its input parameter to the structure of the data.

*Code and data availability.* The synthetic data and code is available at https://github.com/lalcayag/Lidar_filtering. Real data can be found in Simon and Vasiljevic (2018)

*Author contributions.* The author Leonardo Alcayaga implemented the analysis on synthetic and real data and wrote all sections of this manuscript

*Competing interests.* The author delcares not having any competing interest

*Acknowledgements.* I would like to thank Ioanna Karagali for comments and guide through the data from the Østerild Balconies experiments, Robert Menke for the first version of the median-like filter and Ebba Dellwik, Nikola Vasiljevic, Gunner Larsen, Mark Kelly and Jakob Mann for the very useful and clarifying feedback during the analysis and writing process.



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
