# Peer review of "Filtering of pulsed lidars data using spatial information and a clustering algorithm"

_Atmospheric Measurement Techniques, 2019_

## Referee Comment (RC1) · Anonymous Referee #1 · 20 Mar 2020

This paper presents alternative wind lidar data recovery methods, over the traditional carrier-to-noise ratio. The paper presents both a clustering technique and a median-like filter, and evaluates results on both synthetic and real lidar data.

While the paper includes some important results, the presentation is a little clumsy, and I feel the paper could be greatly improved. There needs to be general improvements to the usage of English throughout, examples of which I have highlighted below. The paper overall reads as if several authors composed different sections, there is a lot of repetition of the discussion, and the figures do not flow nicely. While some scrolling/page turning is expected, referring to figure 10 on page 7 requires the reader to turn to page 19. Perhaps there is an alternative way to make your point on page 7? Figure 7 also does not seem to be referred to in the text?

[Figure]

The point I would like to make most clearly is your conclusion states the clustering filter performs best in both synthetic and real data, and increases data availability between 22% and 38%, while also reducing erroneous measurements between 70% and 80%. This is a significant result, and I feel you could make more of this in the paper. There is a lot of discussion on methods used, sometimes repeated several times, but I feel comparatively little on your major results.

Improving the flow of the paper, and removing some of the repeated discussion to focus more on results will greatly enhance your paper.

Minor comments:

Title should read "lidar" rather than "lidars"

Page 1 Line 13 – replace "its adoption" with "their adoption" or similar Line 14/15/16 – the meaning of the sentence beginning "Their capability to measure…." is unclear. Do you mean a single lidar can scan a spatial domain of comparable size to a wind farm? If so, it would be helpful to include an indication on the actual size of a windfarm. By "their increasing accuracy" do you mean increased accuracy over meteorological masts? Line 17 – please be more specific with "traditional wind measurement techniques", for example wind profiling radars can also be used, and are also susceptible to atmospheric conditions. What is "traditional"? Line 18 – please define "lack of references", do you mean a second instrument to compare wind values to? Line 25 – please define VLOS the first time you use it, rather than the second

Page 2 Line 26 – remove the "of" in "between of line-of-sight…." Line 39 – you don't need both "like" and "e.g." together Line 39 – please consider rephrasing the sentence beginning "Complementing all these features….". The sentence is very long and difficult to follow. Line 45 - "….which are capable of classify large data sets…." needs to be reworded for correct English Line 54 – swap the order of "defines" and "always" to read "which always defines a unique…." Line 56 – please define/introduce DBSCAN here, rather than on page 12 Line 58 - "….capable of identify clusters…." should read

"....capable of identifying clusters...."

Page 3 Line 72 – what do you mean by "the wind speed data covers a large horizontal area"? Do you mean you wish to measure winds across a large area? Line 88 – I'm not sure I follow what a "wrong observation" is, as compared to an outlier?

Page 5 Line 99 – change "generate" to "generates" Line 102 – change "make" to "mean" or similar

Page 7 Figure 2 caption – line 3, I believe should read "next" not "nest" Line 149 - "radial" is miss-spelled Line 158 - "en" should be "in"

Page 9 Line 184 - "2" should read "section 2" as done previously Line 189 – the sentence beginning "The noisy areas show...." is very long and hard to follow. Please consider rewording.

Page 10 Line 200 to 203 – these 2 sentences seem to be a repeat of the introduction?

Page 11 Line 229 - "non" should read "not"

Page 12 Line 240 – similar to the comment above, page 10 lines 200 – 203, this section appears to be a repeat of earlier discussions

Page 15 Line 298 – I think you mean "noisy" not "nosy"

Referral to figure 7?

Page 16 Lines 315 to 320 – sentence beginning "This allows us to define...." is very long and difficult to follow Line 320 - "this metrics" should read "these metrics"

Page 18 Line 344 – I think you are missing "are" in "....that two realizations from the same distribution...." Line 365 – should read "....on the other hand...." rather than "in"

Is there a reason why you can't do the same tests to the synthetic data as you are for the real data?

Page 19 Line 372 – remove the second "then" from "....then becomes relevant then...."

Page 20 Line 387 – remove the comma after "both" to read "….in both noisy and reliable…." Line 390 – reverse the order of "be then" to read "then be" Line 391 – replace "its" with "their" to read "….distant from their previous location…." Line 401 – remove "be" and change "benefited" to "benefit" to read "….filter will benefit by…." Line 403 – add "to" to read "….dimensions to the data description."

Page 21 Line 406 – remove "a" to read "….of good measurements…."

I don't get the comparison to synthetic data. You site the advantages of using synthetic data are you know where the noise is, yet you don't have plots showing a comparison to the known noise is?

Page 27 Line 483 – replace "This" with "These" to read "These possible deviations…."

---

## Referee Comment (RC2) · Anonymous Referee #2 · 12 Apr 2020

Alcayaga presents a study about filtering methods for Doppler wind lidar measurements. A new method based on data clustering is developed and compared against the classical CNR filter and a median filter which has become more popular recently. The method is tested in a simulation with artificial turbulence and noise as well as in a real experiment. I think the method is promising and the results that are shown look very interesting. However the manuscript is way too long, not prepared very well and should be rewritten in a much more concise way. The structure currently is confusing with many repetitions and lengthy explanations of minor details, but important information about the data, the methods and the results are missing. Since the topic of the study is relevant and the methods and results could be interesting for the scientific community I would like to see a major revision of the manuscript before it could be

reconsidered for publication in Wind Energy Science. I give general comments about each section as well as specific comments in the following.

**0.1 General comments**

- It has not been shown convincingly that the generated noise in the lidar simulation is realistic and the analysis of the filter in the simulation can thus be considered relevant for real-world measurements.

- The math of the methods is not presented clearly in equations, especially regarding the filters.

- The work is not referencing important work in the field of lidar simulation and daa filtering adequately.

- Section 3.2: Lidar simulators are not new and similar work can be referenced (e.g. Stawiarski et al. 2013, Gasch et al. 2020). Based on these works, the description of the technology could be siginifically shortened. The most important points like the resolution of the synthetic data that is used should be highlighted in a concise way.

- Section 3.3: The noise generation is described with many words and steps that are very hard to follow and confusing. I think it should be possible to describe a noise filter transfer function with a concise mathematical expression. I also think that in this section the characteristics of the synthetic noise should be compared to what is expected from real lidar measurements. Could you for example show a PDF from real measurements of only low CNR data in comparison to the artificial noise? Without any information on how realistic the synthetic noise is, it is hard to judge the quality of the filter from the simulation results.

[Figure]

- Section 4.1 is partly a repetition of things that have been said in the introduction and since CNR-filters are very easy and well known, I think this could be cut much shorter.

- Section 4.2 is supposed to describe the median filter, but does not give the most important parameters. The median of what database is used? Just single scans, multiple scans, the whole scan or just parts of it. Again, I recommend to put the filter description into one or two equations, which would describe it in the best concise way. Menke et al. 2019 and Menke 2020 (dissertation) introduced a modified three-stage median filter for spatial scans. How does the method applied here relate to that?

- Section 4.3 gives a lengthy description of the clustering algorithm, but misses the most important point. Where is the connection between the lidar parameters CNR, Vlos etc and the filtering algorithm. Please give the filter functions for the concrete problem of lidar signals. What is the k-distance function fo the lidar measurement? How is the data sorted in Figurer 8? I doubt that any lidar user can reproduce this method with the information which is given in this section.

- Section 5.1: The author introduces many performance metrics here, of which many are not very useful in my opinion and only add to the confusion of the reader. To me, the interesting metrics are the fraction of good observations (here: $\eta_{recov}$) and the false positive rate (i.e. the percentage of data points that are considered good observations although they are contaminated by noise).

- Section 5.2: I would advice the author to focus on just one most appropriate metric for the analysis of the similarity of the PDFs, especially since the qualitative results are the same and differences between the two metrics are not discussed in Section 6 and 7.

- Section 6.1: I think the the line-of-sight threshold should be discussed in Section

4.2 and not here. What I miss in this section is a plot of the actual LOS velocity fields recovered with the two filters. Lines 403ff give a discussion that is partly repeated in Section 7.1 and should be removed here.

- In section 6.2 the author argues a lot with data recovery, which is not a good metric, because without any filter, the data recovery is perfect, but includes a lot of bad data. The author should focus on the metrics introduced in section 5.2, which is a good choice and the best that can be done. So, I wonder if Figures 15-17 and Table 5 are really useful for the study. One idea would be to replace Figure 16 with a plot of the PDF of the area around the hard target only, comparing the three filters and the original data. Same as for data in different distances to the lidar.

- I think the title "performance assessment" of sections 7.1 and 7.2 is misleading, because those sections mostly evaluate the flaws of the test cases. The performance of the filters is already assessed in the results section.

- Section 7.3 and 8 could probably be combined.

0.2   Specific comments

- p.1, l.1: simultaneous multi-point observations are possible with masts if multiple sonics are installed.

- p.1, l.2: write "lower" instead of "reduced"

- p.1, l.4: "reduced data recovery" compared to what? I am also not sure if "data recovery is the proper term.

- p.1., l.6: "...spatial position, and $V_{LOS}$ smoothness". The abstract needs to be understood without reading the whole manuscript. It is not clear at this point what is meant by spatial position and smoothness.

- p.1,l.13: "its adoption" - "their acceptance"!?

- p.1.,l.21: Since the CNR thresholds are so divers and depend on the conditions and instruments I recommend to not give numbers here.

- p.2.l.37: typo "approaches"

- p.2,l.56: "DBSCAN" acronym should be explained here.

- p.3,l.80: Why are the scanning patterns coherent?

- p.5,l.105: The term "numerical lidar" is very unusual and irritating. I would recommend "lidar simulator" or "virtual lidar".

- p.5,l.112: What does "coarse" mean here? Numbers should be given.

- p.6,Eq.2: The variable names are somewhat confusing, because what is here $\Delta p$ is $\Delta R$ in the references of Smalikho and Banakh and $\Delta p$ in the references is $r_p$ here.

- p.6,l.129: "corresponding range gate center"!

- p.6,l.130: "range gate length" is not very specific. If you give the explanation of $r_p$ from FWHM, you could also give the explanation of $\Delta p$ from the time window of the FFT.

- p.7,l.149: typo "radial"

- p.7,l.154: referencing Figure 10 which is introduced much later, is bad style.

- p.7,l.158: type "in"

- p.8,l.177: again, a figure (Figure 5) is referenced before its introduction.

- p.9,l.180: "The fraction of beams contaminated at each band..."

- p.9,l.183: typo "from".

- p.10,l.201: I do not think you can really give a common value for CNR values. They depend strongly on instruments and location.

- p.10,l.215f: put citation Huang et al in parantheses.

- p.13,l.251: the $m$ in "$m$-dimensional" is not explained.

- p.14,l.285: How does $d_k(n)$ look like for the lidar signal problem?

- p.15,l.298: typo: "noisy"

- p.15,l.298: Figure 9 is referenced before introduction.

- p.15,l.302: Equation 6 is referenced before it appears. Please introduce it before.

- Figure 8b) seems to be moreless the same as Figure 5b.

- p.17,l.333: "PDF" should be in capital letters as an abbreviation.

- p.18,l.344: Something is wrong with the grammer in this sentence.

- p.18,l.345: What is the value of $\alpha$ that is used in this study?.

- p.18,l.345: Again, grammar.

- p.18,l.345f: The numbers about the amount of data that was analyzed should be given in Section 2.

- p.19,l.372: remove one "then".

- p.20,l.386: typo: "account"

- Figure 11: I think this figure is not neccessary. If it is still shown, labels have to be larger.

- Figure 14: typo, should be "phase 2"

- Figure 15: Why is no upper threshold for the CNR filter applied, which would remove the wind turbine hard target from the recovered data?

- Figure 16: I think this plot is not neccessary.

- p.23,l.439: What is meant by "quality of the data"? Probably you mean a lower false positive rate, but how do you know?

- p.23,l.443: Metrics are introduced in Sect. 5.1.

- p.23,l.443f: Again, quality is undefined.

- p.27,l.502: typo: "from"
* * *

---

## Author Comment (AC1) · 19 Jun 2020

Dear reviewer, I highly appreciate your feedback. It helped to greatly improve the manuscript. In the attachment you will find answers to your comments and suggestions in detail.

Please also note the supplement to this comment: https://www.atmos-meas-tech-discuss.net/amt-2019-450/amt-2019-450-AC1-supplement.pdf

---

## Author Comment (AC2) · 19 Jun 2020

Dear reviewer, I highly appreciate your comments, suggestions and the time spent reading the manuscript. They comments are very helpful to improve the manuscript in many aspects.

Please also note the supplement to this comment:
https://www.atmos-meas-tech-discuss.net/amt-2019-450/amt-2019-450-AC2-supplement.pdf

---

## Author Response (AR1)

Comments on "Filtering of pulsed lidars data using spatial information and a clustering algorithm"

**Anonymous Referee #1**

General comments

This paper presents alternative wind lidar data recovery methods, over the traditional carrier-to-noise ratio. The paper presents both a clustering technique and a median-like filter, and evaluates results on both synthetic and real lidar data.

While the paper includes some important results, the presentation is a little clumsy, and I feel the paper could be greatly improved. There needs to be general improvements to the usage of English throughout, examples of which I have highlighted below. The paper overall reads as if several authors composed different sections, there is a lot of repetition of the discussion, and the figures do not flow nicely. While some scrolling/page turning is expected, referring to figure 10 on page 7 requires the reader to turn to page 19. Perhaps there is an alternative way to make your point on page 7? Figure 7 also does not seem to be referred to in the text?

A: The paper will be improved to correct the problems suggested by the reviewer.

The point I would like to make most clearly is your conclusion states the clustering filter performs best in both synthetic and real data, and increases data availability between 22% and 38%, while also reducing erroneous measurements between 70% and 80%. This is a significant result, and I feel you could make more of this in the paper. There is a lot of discussion on methods used, sometimes repeated several times, but I feel comparatively little on your major results. Improving the flow of the paper, and removing some of the repeated discussion to focus more on results will greatly enhance your paper.

A: This is an important comment and will be reflected in the corrected version.

Specific comments

1) Title should read "lidar" rather than "lidars"
   A: The work presents results from a real and a lidar simulator, this is the reason behind lidars instead of lidar.
2) Page 1 Line 13 – replace "its adoption" with "their adoption" or similar Line 14/15/16 – the meaning of the sentence beginning "Their capability to measure: : :." is unclear. Do you mean a single lidar can scan a spatial domain of comparable size to a wind farm? If so, it would be helpful to include an indication on the actual size of a windfarm By "their increasing accuracy" do you mean increased accuracy over meteorological masts? Line 17 – please be more specific with "traditional wind measurement techniques", for example wind profiling radars can also be used, and are also susceptible to atmospheric conditions. What is "traditional"? Line 18 – please define "lack of references", do you mean a second instrument to compare wind values to? Line 25 – please define VLOS the first time you use it, rather than the second
   A:
   - Page 1 Line 13. Corrected.
   - Page 1 Line 14/15/16. Since wind farm vary in installed capacity and size, giving one number it is not easy. But spacing of large turbines can be in the order of a kilometer (assuming 6D streamwise spacing for turbines of around 150m rotor diameter, the long-range scanners here can cover up to 7km x 10km, meaning several turbines), for clarity, the

sentence was deleted. The technology has developed the last years to increase the laser energy and the backscatter signal quality.

- Page 1 Line 17. The sentence was corrected to refer to meteorological masts.

- Page 1 Line 18. Yes, clarified in the text.

- Page 1 Line 25. Corrected.

3) Page 2 Line 26 – remove the "of" in "between of line-of-sight: : :." Line 39 – you don't need both "like" and "e.g." together Line 39 – please consider rephrasing the sentence beginning "Complementing all these features: : :.". The sentence is very long and difficult to follow. Line 45 - ": : :.which are capable of classify large data sets: : :." needs to be reworded for correct English Line 54 – swap the order of "defines" and "always" to read "which always defines a unique: : :." Line 56 – please define/introduce DBSCAN here, rather than on page 12 Line 58 - ": : :.capable of identify clusters: : :." should read ": : :.capable of identifying clusters: : :."

A:

- Page 2 Line 26. Corrected.

- Page 2 Line 39. Corrected.

- Page 2 Line 45. Corrected.

- Page 2 Line 54. Corrected.

- Page 2 Line 56. Corrected.

- Page 2 Line 58. To keep the introduction section short, DBSCAN definition needs its own section, this is mentioned in the corrected version.

- Page 2 Line 58. Corrected.

4) Page 3 Line 72 – what do you mean by "the wind speed data covers a large horizontal area"? Do you mean you wish to measure winds across a large area? Line 88 – I'm not sure I follow what a "wrong observation" is, as compared to an outlier?

A:

- Page 3 Line 72. It will be rephrased for clarity.

- Page 3 Line 88. It will be rephrased for clarity.

5) Page 5 Line 99 – change "generate" to "generates" Line 102 – change "make" to "mean" or similar.

A:

- Page 5 Line 99. It will be corrected.

- Page 5 Line 102. It will be corrected.

6) Page 7 Figure 2 caption – line 3, I believe should read "next" not "nest" Line 149 - "radial" is miss-spelled Line 158 - "en" should be "in".

A:

- Page 7 Figure 2 caption. It will be corrected.

- Page 7 Line 149. It will be corrected.

- Page 7 Line 158. It will be corrected.

7) Page 9 Line 184 - "2" should read "section 2" as done previously Line 189 – the sentence beginning "The noisy areas show: : :." is very long and hard to follow. Please consider rewording.
A:

- Page 9 Line 184. It will be corrected.

- Page 9 Line 189. It will be rephrased for clarity.

8) Page 10 Line 200 to 203 – these 2 sentences seem to be a repeat of the introduction?
A:

- Page 10 Line 200 to 203. Sentences It will be eliminated.

9) Page 11 Line 229 - "non" should read "not"
A:

- Page 11 Line 229. It will be corrected.

10) Page 12 Line 240 – similar to the comment above, page 10 lines 200 – 203, this section appears to be a repeat of earlier discussions
A:

- Page 12 Line 240. It will be rephrased.

11) Page 15 Line 298 – I think you mean "noisy" not "nosy"
A:

- Page 15 Line 298. It will be corrected.

12) Referral to figure 7?
A:

- Referral was only in the caption of Figure  8, which is complementary. It will be corrected in the text.

13) Page 16 Lines 315 to 320 – sentence beginning "This allows us to define: : :." is very long and difficult to follow Line 320 - "this metrics" should read "these metrics"
A:
-Page 16 Line 315 to 320. It will be rephrased.

14) Page 18 Line 344 – I think you are missing "are" in ": : :.that two realizations from the same distribution: : :." Line 365 – should read ": : :.on the other hand: : :." rather than "in"
A:
-Page 18 Line 344 and 365. It will be corrected.

15) Is there a reason why you can't do the same tests to the synthetic data as you are for the real data?
A: It is possible, but tests on real data are based on reliable observations on a range of CNR values, due to the lack of references available, which is not the case for synthetic data. In the ideal case, the test applied on synthetic data would be the best for real data.

16) Page 19 Line 372 – remove the second "then" from ": : :.then becomes relevant then: : :."
A:

-Page 19 Line 372. It will be corrected.

17) Page 20 Line 387 – remove the comma after "both" to read ": : :.in both noisy and reliable: : :." Line 390 – reverse the order of "be then" to read "then be" Line 391 – replace "its" with "their" to read ": : :.distant from their previous location: : :." Line 401 – remove "be" and change "benefited" to "benefit" to read ": : :.filter will benefit by: : :." Line 403 – add "to" to read ": : :.dimensions to the data description."
A:
- Page 20 Lines 387, 390, 391 and 401. It will be corrected.

18) Page 21 Line 406 – remove "a" to read ": : :.of good measurements: : :." I don't get the comparison to synthetic data. You site the advantages of using synthetic data are you know where the noise is, yet you don't have plots showing a comparison to the known noise is?
A:

- Page 21 Line 406. It will be corrected. The position of the noise for an individual scan is shown in Figure 3 (c).

19) Page 27 Line 483 – replace "This" with "These" to read "These possible deviations: : :."
A:

- Page 27 Line 483. It will be corrected.

**Anonymous Referee #2**

Alcayaga presents a study about filtering methods for Doppler wind lidar measurements.A new method based on data clustering is developed and compared against the classical CNR filter and a median filter which has become more popular recently. The method is tested in a simulation with artificial turbulence and noise as well as in a real experiment. I think the method is promising and the results that are shown look very interesting. However the manuscript is way too long, not prepared very well and should be rewritten in a much more concise way. The structure currently is confusing with many repetitions and lengthy explanations of minor details, but important information about the data, the methods and the results are missing. Since the topic of the study is relevant and the methods and results could be interesting for the scientific community I would like to see a major revision of the manuscript before it could be reconsidered for publication in Wind Energy Science. I give general comments about each section as well as specific comments in the following.

General Comments

1) It has not been shown convincingly that the generated noise in the lidar simulation is realistic and the analysis of the filter in the simulation can thus be considered relevant for real-world measurements.
A: The procedural noise implemented here aims to generate V_LOS values smoother than the ones observed at very low CNR, and closer to "reasonable" V_LOS. Figure 1 below shows the distributions of synthetic, contaminated V_LOS values and real V_LOS data with CNR values below -32dB. From this Figure it is possible to see that the synthetic noise generates V_los closer to reliable values and thinner tails. The consequence of this is a more subtle contamination, which is harder to detect by the filters presented in this work. Additionally, the principle of coherent noise is to generate areas of contamination that are smoother in space, which also makes more difficult differentiate contaminated observations and clean data via the distance generated by $\Delta$V_LOS and for DBSCAN, and a fixed threshold for the median-like filter. In summary, the intention of this implementation is not to recreate real noise (its nature is relatively unknown), but to test the filter in harder conditions than real situations.

[Figure]

Figure 1: Pdf of V_LOS for contaminated synthetic data (one mean wind speed direction) and real data from the Balconies experiment at 200 m.a.g.l.

2) The math of the methods is not presented clearly in equations, especially regarding the filters.

A: The iterative operation of DBSCAN on discrete data is non-linear and is defined algorithmically. To the best of my knowledge, there are not references of transfer functions or reduced mathematical expressions of its frequency response for instance. Regarding the median-like filter, as mentioned in the paper, its most obvious parallel is the median filter used in image processing. As DBSCAN, this filter is non-linear, and it lacks a defininition in frequency domain as a transfer function. The development of theoretical expressions in this sense is out of the scope of this work. ### This is stated in the corrected version of the paper.

3) The work is not referencing important work in the field of lidar simulation and data filtering adequately.
   A: The suggested references will be checked and included accordingly.

4) Section 3.2: Lidar simulators are not new and similar work can be referenced (e.g. Stawiarski et al. 2013, Gasch et al. 2020). Based on these works, the description of the technology could be siginificantly shortened. The most important points like the resolution of the synthetic data that is used should be highlighted in a concise way.
   A: The references mentioned were considered and the description of the simulator will be rewritten in a more concise way.

5) Section 3.3: The noise generation is described with many words and steps that are very hard to follow and confusing. I think it should be possible to describe a noise filter transfer function with a concise mathematical expression. I also think that in this section the characteristics of the synthetic noise should be compared to what is expected from real lidar measurements. Could you for example show a PDF from real measurements of only low CNR data in comparison to the artificial noise? Without any information on how realistic the synthetic noise is, it is hard to judge the quality of the filter from the simulation results.
   A: The coherent noise implemented here is not linear and is defined, as a parallel to DBSCAN, algorithmically. There is not a clear transfer function that defines it a priori and it can be better described by the V_LOS distribution after contamination. Figure 1 will be included in the corrected version.

6) Section 4.1 is partly a repetition of things that have been said in the introduction and since CNR-filters are very easy and well known, I think this could be cut much shorter.
   A: Section will be reworded in a more concise way.

7) Section 4.2 is supposed to describe the median filter, but does not give the most important parameters. The median of what database is used? Just single scans, multiple scans, the whole scan or just parts of it. Again, I recommend to put the filter description into one or two equations, which would describe it in the best concise way. Menke et al. 2019 and Menke 2020 (dissertation) introduced a modified three-stage median filter for spatial scans. How does the method applied here relate to that?
   A: As mentioned earlier, the non-linear median-filter is defined algorithmically, not via equations. The filter is applied on single scans (it is a filter that operates spatially), and this is clarified in the corrected text. The filter closely related to the one described by Menke (2019), since it uses a moving window in the laser beam direction, the first stage. Nevertheless It does not applies a global filtering stage, which is replaced by a second moving window in the azimuth direction. The paragraph was reviewed and corrected to make it more clear.

8) Section 4.3 gives a lengthy description of the clustering algorithm, but misses the most important point. Where is the connection between the lidar parameters CNR, Vlos etc and the filtering algorithm. Please give the filter functions for the concrete problem of lidar

signals. What is the k-distance function fo the lidar measurement? How is the data sorted in Figurer 8? I doubt that any lidar user can reproduce this method with the information which is given in this section.

A: As mentioned in 2), there is not a mathematical expression in the form of a transfer function for DBSCAN, since it is defined algorithmically. The connection between lidar parameters, or features, and DBSCAN is the definition of the observational space, where all (Euclidian) distances are calculated. This is better explained in the corrected text, as well as the k-distance function, or the function of the distance of each point to its k-nearest neighbor.

9) Section 5.1: The author introduces many performance metrics here, of which many are not very useful in my opinion and only add to the confusion of the reader. To me, the interesting metrics are the fraction of good observations (here: $\square$recov) and the false positive rate (i.e. the percentage of data points that are considered good observations although they are contaminated by noise).

A: The false positive rate (false negative in the work, positive is noise detection) is equivalent to the fraction of noise detected, but it does not consider information of the fraction of contaminated observations in the scan. High recovery rates with low false positive rate (negative) might be only a low fraction of noise. This is the reason to include also a metric that takes into account the noise fraction.

10) Section 5.2: I would advice the author to focus on just one most appropriate metric for the analysis of the similarity of the PDFs, especially since the qualitative results are the same and differences between the two metrics are not discussed in Section 6 and 7.

A: I agree with this comment. The result of one of them will be only mentioned for a fair comparison in the corrected text.

11) Section 6.1: I think the line-of-sight threshold should be discussed in Section 4.2 and not here. What I miss in this section is a plot of the actual LOS velocity fields recovered with the two filters. Lines 403ff give a discussion that is partly repeated in Section 7.1 and should be removed here.

A: Since the results of synthetic scans are presented here, discussion on V_LOS threshold is better pictured fter the filters are applied. The discussion will be initiated in 4.2, and a figure comparing the two filters will be included in the corrected text.

12) In section 6.2 the author argues a lot with data recovery, which is not a good metric, because without any filter, the data recovery is perfect, but includes a lot of bad data. The author should focus on the metrics introduced in section 5.2, which is a good choice and the best that can be done. So, I wonder if Figures 15-17 and Table 5 are really useful for the study. One idea would be to replace Figure 16 with a plot of the PDF of the area around the hard target only, comparing the three filters and the original data. Same as for data in different distances to the lidar.

A: Data recovery is very important in this work indeed, since the main motivation to explore a different filtering technique is to increase the amount of data available, which can be very poor when we use the most conservative CNR threshold. However, as the referee suggest, this is worthless if data quality is bad. This is the reason to complement the performance assessment with the metrics in 5.2. Figure 16 will be modified to consider this suggestion.

13) I think the title "performance assessment" of sections 7.1 and 7.2 is misleading, because those sections mostly evaluate the flaws of the test cases. The performance of the filters is already assessed in the results section.

A: A change in the section title will be made to clarify its intention.

14) Section 7.3 and 8 could probably be combined.

A: Even though section 7.3 give some final remarks it still discuss on computational performance of the clustering filter and possible imporvements. The section will be revised to make this more clear.

Specific comments

1) p.1, l.1: simultaneous multi-point observations are possible with masts if multiple sonics are installed.
   A: Indeed this is possible, at a high cost though, and it is not very common for wind resource assessment for instance. That is the meaning of the sentence.

2) p.1, l.2: write "lower" instead of "reduced"
   A: It will be corrected in the text.

3) p.1, l.4: "reduced data recovery" compared to what? I am also not sure if "data recovery is the proper term.
   A: The reduced data recovery is compared to the total amount of data available.

4) p.1., l.6: "...spatial position, and VLOS smoothness". The abstract needs to be understood without reading the whole manuscript. It is not clear at this point what is meant by spatial position and smoothness.
   A: It will be rephrased for clarity.

5) p.1,l.13: "its adoption" - "their acceptance"!?
   A: It will be corrected

6) p.1.,l.21: Since the CNR thresholds are so divers and depend on the conditions and instruments I recommend to not give numbers here.
   A: It will be modified for clarity. The intention is to show values used in the reference cited.

7) p.2.l.37: typo "approaches"
   A: It will be corrected.

8) p.2,l.56: "DBSCAN" acronym should be explained here.
   A: It will be corrected

9) p.3,l.80: Why are the scanning patterns coherent?
   A: It is intented to mean meaningful

10) p.5,l.105: The term "numerical lidar" is very unusual and irritating. I would recommend "lidar simulator" or "virtual lidar".
    A: The term was already used in Meyer (2017) but the suggestion will be considered.

11) p.5,l.112: What does "coarse" mean here? Numbers should be given.
    A: Coarser means a grid of range gates and beam spacing that is actually much coarser than the sampling spacing frequency of the lidar, and represents the spatial and time averaging of the instrument. The numbers of this are in table 3. The term will be clarified in the corrected text.

12) p.6,Eq.2: The variable names are somewhat confusing, because what is here $\Delta p$ is $\Delta R$ in the references of Smalikho and Banakh and $\Delta p$ in the references is rp here.
    A: The intention was not to use the same notation as Smalikho and Banakh but Meyer (2017)

13) p.6,l.129: "corresponding range gate center"!
    A: it will be corrected

14) p.6,l.130: "range gate length" is not very specific. If you give the explanation of rp from FWHM, you could also give the explanation of $\Delta p$ from the time window of the FFT.
    A: Since this terms are well known, their definition was done in simpler terms.

15) p.7,l.149: typo "radial"
    A: It will be corrected

16) p.7,l.154: referencing Figure 10 which is introduced much later, is bad style.

A: It will be corrected

17) p.7,l.158: type "in"

A: It will be corrected

18) p.8,l.177: again, a figure (Figure 5) is referenced before its introduction.

A: It will be corrected

19) p.9,l.180: "The fraction of beams contaminated at each band..."

A: It will be corrected

20) p.9,l.183: typo "from".

A: It will be corrected

21) p.10,l.201: I do not think you can really give a common value for CNR values. They depend strongly on instruments and location.

A: This will be removed in the corrected version.

22) p.10,l.215f: put citation Huang et al in parantheses.

A: It will be corrected

23) p.13,l.251: the m in "m-dimensional" is not explained.

A: It will be explained

24) p.14,l.285: How does dk(n) look like for the lidar signal problem?

A: It is shown in figure 9. This figure will be probably removed in the final version.

25) p.15,l.298: typo: "noisy"

A: It will be corrected

26) p.15,l.298: Figure 9 is referenced before introduction.

27) A: It will be corrected

28) p.15,l.302: Equation 6 is referenced before it appears. Please introduce it before.

A: It will be corrected

29) Figure 8b) seems to be moreless the same as Figure 5b.

A: They come from bad and good scans respectively. It will be probably removed in the corrected version.

30) p.17,l.333: "PDF" should be in capital letters as an abbreviation.

A: It will be corrected

31) p.18,l.344: Something is wrong with the grammer in this sentence.

A: It will be corrected, an "are" was missing.

32) p.18,l.345: What is the value of that is used in this study?.

A: The value of alpha is 0.05

33) p.18,l.345: Again, grammar.

A: It will be corrected

34) p.18,l.345f: The numbers about the amount of data that was analyzed should be given in Section 2.

35) A: It will be corrected

36) p.19,l.372: remove one "then".

37) A: It will be corrected

38) p.20,l.386: typo: "account"

A: It will be corrected

39) Figure 11: I think this figure is not neccessary. If it is still shown, labels have to be larger.

A: I agree and it will be removed in the corrected version.

40) Figure 14: typo, should be "phase 2"

41) A: It will be corrected

42) Figure 15: Why is no upper threshold for the CNR filter applied, which would remove the wind turbine hard target from the recovered data?

A: It is applied indeed, this will be written more clear.

43) Figure 16: I think this plot is not neccessary.

A: This comment will be considered.

44) p.23,l.439: What is meant by "quality of the data"? Probably you mean a lower false positive rate, but how do you know?

A: It refers mostly to extreme values when compared to more reliable CNR data distribution.

45) p.23,l.443: Metrics are introduced in Sect. 5.1.

A: It will be corrected

46) p.23,l.443f: Again, quality is undefined.

A: It will be better explained

47) p.27,l.502: typo: "from"

A: It will be corrected

List of changes

1) Changes (rewording) in the introduction and data description in section 2.
2) Changes in Figures suggested by reviewers. Figures 3 (now 2), 4 (now 3), 5 (now 4) and 16 (now 13) were modified.
3) Figures 7, 8 and 9 are replaced by Figure 6.
4) Figure 11 is now A1 in appendix A.
5) Changes in Sections 3 and 4 regarding methodology description (Lidar simulator in 3.2, noise generation 3.3 and filters description in 4.1, 4.2 and 4.2) accounts for comments from reviewers.
6) Section 6.1 was modified and results in Figure 8, suggested by Anonymous Reviewer # 2, included.
7) Discussion in sections 7.1, 7.2 and 7.3 were merged in only one section.
8) Part of 7.3 was moved to 8.
9) Part of sections 4.2 and 6.1 were moved to Appendix A.

**Filtering of pulsed lidars data using spatial information and a clustering algorithm**

Leonardo Alcayaga[1]

[1]DTU Wind Energy, Frederiksborgvej 399, 4000 Roskilde, Denmark

**Correspondence:** Leonardo Alcayaga (lalc@dtu.dk)

**Abstract.** Wind lidars present advantages over meteorological masts, including simultaneous multi-point observations, flexibility in measuring geometry, and reduced installation cost; but wind lidars come with the 'cost' of increased complexity in terms of data quality and analysis. Carrier-to-noise ratio (CNR) has been the metric most commonly-used to recover reliable observations from lidar measurements, but with severely reduced data recovery. In this work we apply a clustering technique to identify unreliable measurements from pulsed lidars scanning a horizontal plane, taking advantage of all data available from the lidars—not only CNR, but also line-of-sight wind speed ($V_{LOS}$), spatial position, and $V_{LOS}$ smoothness. The performance of this data filtering technique is evaluated in terms of data recovery and data quality, against both a median-like filter and a pure CNR-threshold filter. The results show that the clustering filter is capable of recovering more reliable data in noisy regions of the scans, increasing the data recovery up to 38% and reducing by at least two thirds the acceptance of unreliable measurements, relative to the commonly used CNR-threshold. Along with this, the need for user intervention in the setup of data filtering is reduced considerably, which is a step towards a more automated and robust filter.

**1 Introduction**

Long range scanning wind lidars are useful tools, and  their adoption has grown rapidly in recent years in wind energy applications (Vasiljevic et al., 2016).  Scanning wind lidars can measure time evolution and spatial characteristics of wind fields over large domains, at a lower cost of installation  than meteorological masts.  Nevertheless, atmospheric conditions and instrument noise  can have an important impact on the data quality. For long-range scanning lidars  this becomes an important issue due to the lack of  additional instruments placed over the measurement area that would be useful to compare data quality, since noise can contaminate large portions of the scanning domain. The most commonly used criteria to retrieve reliable observations is a threshold on values of the Carrier-to-noise ratio, CNR,  threshold that will depend on site conditions, experimental setup and  the instrument manufacturer (Gryning et al., 2016; Gryning and Floors, 2019).  Despite CNR threshold retrieve quality observations, its application might result

in large amounts of  good data rejected in regions far from the instrument,   where CNR has decreased rapidly with distance. To cope with this issue Meyer Forsting and Troldborg (2016) and Vasiljević et al. (2017) have proposed filters based on the smoothness and continuity of the wind field. Such filters work by detecting discrete or anomalous steps   (above a certain threshold, predefined by

30 the user) in line-of-sight wind speed, $V_{LOS}$,  compared to its local (moving) median . Beck and Kühn (2017) first and Karagali et al. (2018) in an adapted version, follow a different approach (here called KDE filter, from Kernel Density Estimate) based on the statistical self-similarity of the data, which, in simple terms,  means that reliable observations are alike and will be located close together in the observational space. The probability density distribution of observations (estimated via KDE) in a dynamically normalized $V_{LOS} - CNR$ space shows that measurements

35 likely to be valid are located in a high data density region. Observations sparsely distributed beyond a boundary defined by a threshold in the acceptance ratio, or the ratio between the probability density of any observation and the maximum probability density over the whole set of measurements, are finally identified as noise. Both approaches need the definition of one or more thresholds and a window size, either in time for the KDE filter, or in space for the wind field smoothness approach. These parameters are dependent on different characteristics of the data, like the lidar scanning pattern for instance.

40 Both approaches miss important and complementary information, either neglecting the   stregth of the signal back-scattering (quantified by CNR) or the spatial distribution and smoothness of the wind field. Moreover, in both  approaches the position of observations is not taken into account, information that can shed light on areas permanently showing anomalous values of $V_{LOS}$ or CNR, like  hard targets.  Including all these features within the smoothness approach is difficult, since CNR is not a smooth field like $V_{LOS}$.

45 Moreover, considering smoothness and position in the KDE filter   results in a computationally costly kernel density estimation, if we look for an optimal bandwidth parameter in a higher dimensional space  , with a fine resolution of the kernel density estimate.

50 Data self similarity – over any scale in the case of fractals or a range of them in real situations (Mandelbrot, 1983) – is closely related to clustering techniques (Backer, 1995), which  can classify large data sets with many different features at a relatively low computational cost.  The KDE filter approach shares some characteristics with the popular $k\text{-}means$ clustering algorithm MacQueen (1967), since they define one (or several for $k - means$) specific group of data belonging to an unique category (or cluster)  whose size and location on

55 the observational space will depend on data density or, more specifically, on a kernel density estimation. The main difference between these two algorithms is the way they treat sparse data points that fall in low density regions.   Unlike the KDE filter, which rejects noise via the acceptance ratio, $k - means$ assigns sparse points to the cluster with the nearest center, no matter if they are outliers or present unlikely values from a physical point of view.

60

The Density Based Spatial Clustering for Applications with Noise algorithm, or `DBSCAN`  (Ester et al., 1996; Pedregosa et al., 2011), introduced in Section 4.3, presents several advantages over $k\text{-}means$ in detecting clusters in a higher dimensional space:  it introduces the notion of noise/sparsely distributed observations , it does not need prior knowledge of the number of clusters in the data and it is capable of  identifying clusters of arbitrary shape. To the best of our knowledge, this is the first time that this type of clustering algorithm is applied to identify not reliable observations from pulsed lidars. This approach, which can be understood as a natural extension of the KDE filter, is compared to the smoothness based filter on two types of data: synthetic wind fields data as a controlled test case, and real data.

This paper is organized as follows: Section 2 describes the real data used to test the different filtering approaches, and Section 3 presents the synthetic data used during a controlled test as well as the methodology to obtain it. Section 4 then gives a description of the different filters applied in this study to both data sets, to continue with the definition of the performance tests in Section 5. In Section 6 the  performance tests are presented along with a discussion on their validity and quality. Section 7 discuses the quality of the methodology behind the tests and the advantages and disadvantages of the proposed approach. Section 8 presents the conclusions of this study.

**2 Real data: Østerild Balconies experiment**

The filtering techniques presented here were tested on lidar measurements made at the Østerild Test Centre located in northern Jutland, Denmark, see Figure 1.  Known as the Østerild Balconies experiment (Mann et al., 2017; Karagali et al., 2018; Simon and Vasiljevic, 2018), this measuring campaign aimed to characterize horizontal flow patterns above a flat, heterogeneous forested landscape at two heights relevant for wind energy applications. , covering an area of around 50 km$^2$ , and a wide range of scales, both in time and space.

The experiment consist of two measuring phases (see Table 1) with two long-range WindScanners performing Plan Position Indicator (PPI) scanning patterns, aligned in the North-South axis and installed at 50 m a.g.l. during phase 1 and 200 m a.g.l. in phase 2. WindScanners (Vasiljevic et al., 2016) consist of two or more spatially separated lidars which are synchronized to perform coherent scanning patterns, allowing the retrieval of two or three dimensional velocity vectors at  different points in space. These experiments were conducted between April and August of 2016 (Simon and Vasiljevic, 2018). In each

**Table 1.** Characteristics of the Balconies experiment, from Karagali et al. (2018). The scans are not instantaneous neither totally synchronous, with a horizontal sweep speed of 2°/s in the azimuth direction in a range of 90°, with a total time of 45 s per scan.

| Phase | Measurement start | Measurement end |
|---|---|---|
| 50 m a.g.l. (1) | 2016-04-12 12:45:41 | 2016-06-17 12:48:01 |
| 200 m a.g.l. (2) | 2016-06-29 13:35:56 | 2016-08-12 09:09:55 |
| **Scanner** | **Location coordinates, [m]** | **Scanning pattern, west** |
| Southern lidar | 492768.8 (East) 6322832.3 (North) | 344°-256°, 2° steps |
| Northern lidar | 492768.7 (East) 6327082.4 (North) | 196°-284°, 2° steps |

[Figure]

**Figure 1.** (a) Location of the Østerild Turbine Test Center, place of the Balconies experiments, northern Jutland, Denmark (copyright 2009 Esri). (b) Detail of the test center site, with the location of the meteorological masts where north (blue) and south (red) WindScanners were installed. During the measurement campaign the PPI scans pointed both west in some periods and both east in other (copyright 2017 DigitalGlobe, Inc.).

phase, the northern and southern lidars scanned in the West and East direction relative to the corresponding meteorological masts, where they were installed. The data used in this study originated from both phases of the experiment, with PPIs pointing to the west. For more details about the experiment, lidars and terrain characteristics see (Karagali et al., 2018; Vasiljevic et al., 2016; Simon and Vasiljevic, 2018).

This dataset is well suited to test different data filtering techniques. A large measurement area will be affected by local terrain and atmospheric conditions, like clouds or large hard targets. Moreover, at this scale lidars reach their measuring limitations, since the back-scattering from aerosols decrease rapidly with distance (Cariou, 2015).

**3 Synthetic data**

Assessing and comparing the performance of filters is challenging with no reference available to verify that rejected or accepted observations are  reliable or bad observations. This is especially difficult for long-range scanning lidars, since their measurements cover large areas and, due to spatial variability, a valid reference would need several secondary anemometers scattered over the scanning area. Testing filters on a controlled and synthetic data set, contaminated with a well defined noise, presents an option to deal with this problem. In this study, the filters presented in Section 4 are tested on individual scans sampled from synthetic wind fields generated using the Mann turbulence spectral tensor model (Mann, 1994), and contaminated with procedural noise (Perlin, 2001).

**3.1 Synthetic wind fields generation**

Synthetic PPI scans are sampled by a  lidar simulator from synthetic wind fields generated via the Mann-model (Mann, 1998) in a horizontal, two-dimensional square domain of 2048 x 2048 grid points, with dimensions 9200 m x 7000 m. The generated turbulence fields are the result of input parameters of the of turbulence spectral tensor model, namely, length-scale, $L$, turbulence energy dissipation $\alpha\epsilon^{2/3}$, and anisotropy, $\Gamma$. The fields generated correspond to wind speed fluctuations,  to which the desired average wind speed mean is subsequently added. Depending on the initial random seed  used, different wind field realizations with the exact same turbulence statistics can be generated. For details on wind field generation using the the Mann-model, refer to Mann (1998). Table 2 shows the range of values used for the generation of two-dimensional wind fields. Large values of $\alpha\epsilon^{2/3}$ or  small scale turbulence for instance,  mean that sudden spatial changes in wind speed are more likely, which increase the false identification of outliers. Mean wind direction, turbulence anisotropy and length scale will also affect the sampling due to the lidars  measuring characteristics.

**3.2 Lidar simulator**

~~The numerical model of a long-range pulsed lidar attemps to mimic what the real instrument does obtaining, for instance, the data described in Section 2. Even though its implementation is very crude andsimplified, it allows to generate sampled scans with reasonable spatial smoothness and complexity, but it is, however, not able to reproduce either noise nor CNR values. Since the interest of this study is to test filters over a large number of scans, its simplicity allows a quick sampling over high resolution synthetic wind fields , thus reducing computational time~~

Lidar simulators has been presented previously by Stawiarski et al. (2013) and Meyer Forsting and Troldborg (2016). They sample $V_{LOS}$ values from wind fields generated via Large Eddy Simulations (LES), mimicking the operational principle of

**Table 2.** Synthetic wind field characteristics and parameters.

| Parameter | Values |
|---|---|
| $L$, m | 62, 125, 250, 500, 750, 1000 |
| $\alpha\epsilon^{2/3}$, m$^{4/3}$s$^{-2}$ | 0.025, 0.05, 0.075 |
| $\Gamma$ | 0, 1, 2, 2.5, 3.5 |
| Number of seeds used | 10 |
| Mean wind speed, $U$ m/s | 15 |
| Mean wind speed direction range, degrees | 90 to 270 |
| Total number of scans generated | 4305 |

130  lidars by proper time and spatial (probe volume) averaging of the background wind field. The lidar simulator presented here follows the same principles, this time sampling from synthetic wind fields generated via Mann-model. ~~The sequence followed by the model in sampling from the high resolution wind fields is specified below (see also Figure ?? for more details): A coarse two dimensional mesh in polar coordinates $(r, \theta)$ is generated, with radial (range gates) and azimuth ranges and resolution described in Table 3. A set of non overlapping nested meshes are then constructed locally, centered in each point in an upper level, coarser mesh . Each nested mesh has 21 x 51 grid points in the radial and azimuth directions, respectively. The resolution~~

135

The simulator receives scanning pattern characteristics as input (beam range, range gate step, azimuth angles range and azimuth angle steps) to generate a primary mesh with the sampling positions on top of background wind field. Following the

140  measuring principle of the lidar, the $V_{LOS}$ observed at each position in this mesh will represent averages of a continuous along each range gate step (due to probe volume averaging) and an average of many azimuth positions within the azimuth step, due to the almost continuous sweep of the  lidar's beam. The simulator mimics this generating a secondary, refined mesh with $N_r$ points in each range gate and $N_\phi$ beams within each azimuth step. The

145   background wind field components, $U$ and $V$, are then interpolated on this secondary mesh and projected on each refined beam to obtain $V_{LOS}$  using equation (1), with $\theta$ being the corresponding beam azimuth angle.

$$V_{LOS} = \cos(\theta)U + \sin(\theta)V \tag{1}$$

150   The final step is the spatial (probe volume) averaging, and the azimuth (sweeping) averaging around each position in the primary mesh. Spatial averaging is done applying a weighting function on all $V_{LOS}$

$$w = \frac{1}{2\Delta p}\left\{\mathrm{Erf}\left[\frac{(r-F)+\Delta p/2}{r_p}\right] - \mathrm{Erf}\left[\frac{(r-F)-\Delta p/2}{r_p}\right]\right\}$$

along each refined beam. The weighting function used here is defined in equation (2), as in Banakh and Smalikho (1997) and Smalikho and Banakh (2013).  This function will assign weights to each point in the  refined beam according to its distance to the range gate position in the primary mesh, $F$, and the instrument probe volume parameters, namely, range gate length, $\Delta p$, and full width at half maximum, $\Delta l$ (cf. Table 3)

$$\mathrm{Erf}(x) = \frac{2}{\sqrt{\pi}}\int_0^x \exp(-t^2)dt$$

. Here, $\mathrm{Erf}(x)$ is the error function, and

$$r_p = \frac{\Delta l}{2\sqrt{\ln(2)}}$$

 $r_p$ the beam width contribution to the volume averaging.

$$w = \frac{1}{2\Delta p}\left\{\mathrm{Erf}\left[\frac{(r-F)+\Delta p/2}{r_p}\right] - \mathrm{Erf}\left[\frac{(r-F)-\Delta p/2}{r_p}\right]\right\} ; \quad r_p = \frac{\Delta l}{2\sqrt{\ln(2)}} \tag{2}$$

 The azimuth averaging is the arithmetic mean of the $N_\phi$ values of $V_{LOS}$  at each range gate after spatial averaging. It represents the accumulation information of the back-scattered signal spectra as they sweep an azimuth sector  before estimation of the spectral peak and $V_{LOS}$.

~~The local fine polar mesh and averaging in the azimuth direction here play the role of the rapid sweeping of the laser beam, taking into account that the measured $V_{LOS}$ will be the result of the accumulation of information on the back-scattering spectra within the $2°$ azimuth steps, as well as the radial/line averaging around each range gate. This model is, of course, very simplistic and does not contain information about back-scattering spectra nor Doppler effect. Moreover, it does not mimic the non-instantaneous nature of the real scans, but rather assumes that all beams cross the measuring domain simultaneously, recording radial wind speeds at the same time. Nevertheless, it will generate scans with radial speeds projected from a realistic wind field and can be contaminated with noise (presented in Section 3.3) under controlled conditions, which is the final goal of this test.~~

**Table 3.** The characteristics of the lidar simulator and real long-range lidar (Karagali et al., 2018; Vasiljevic et al., 2016) used for the controlled test of the filters.

| | Simulator | Real |
|---|---|---|
| Azimuth range | 256° - 344° | 256° - 344° |
| Azimuth step | 2° | 2° |
| Beam length | 7000 [m] | 7000 [m] |
| Range gate length, $\Delta p$ | 35 [m] | 35 [m] |
| Full width at half maximum, $\Delta l$ | 75 [m] | 75 [m] |
| Sweeping time per scan | Instantaneous | 45 [s] |
| Primary mesh size (radial x azimuth) | 45 x 198 | - |
| Secondary mesh size at each range gate ($N_r$ x $N_\phi$) | 21 x 51 | - |
| Total secondary mesh size ($N_r$ x $N_\phi$) | 21 x 51 | - |

**3.3 Synthetic noise generation**

The most simple noise that can be used to contaminate synthetic scans is sparse, uniformly distributed outliers. This noise, also known as salt and pepper noise, is easily detected and eliminated by median-like filters, when extreme discrete steps affect the smoothness of an image (Huang et al., 1979; Burger and Burge, 2008). Nevertheless, noise in real scans comes as regions of anomalously high and/or low $V_{LOS}$ and they can pass through the filter undetected. Procedural noise, introduced by (Perlin, 2001) to recreate synthetic textures on surfaces for computer graphics applications, creates regions of coherent noise that resembles better the spatial distribution of scanning lidars measurements. For the two-dimensional case, the procedural noise function $N(x,y)$ maps two-dimensional coordinates, $(x,y)$, onto the range $[-1,1]$ as follows,

- A two-dimensional grid of $m$ by $n$ elements is generated, and a pseudo-random, two-dimensional unit gradient, $\mathbf{g}_{ij} = (g_x, g_y)$, is assigned to each grid point

$(x_i, y_j)$. The pseudo-randomness  rises from the fact that $\mathbf{g}_{ij}$ are picked from a pre-computed list of gradients with length $l << m \times n$. We select values from this list using the index permutation grid $p_{ij} \in \{0,...,l\}$ also with $m \times n$ elements. Then, $\mathbf{g}_{ij}$ will correspond to the gradient in the position $p_{ij}$ of the  pre-computed list. Elements $p_{ij}$ are shuffled for each realization.

-

- ~~To estimate the noise level of each point $p$, the contribution from each of the nearest pseudo-random gradients is calculated as the dot product between the corresponding gradient and the distance vector , $\mathbf{d}$, from the gradient position to $p$. Finally, each contribution is added directly after weighting by the inverse of the magnitude of $\mathbf{d}$, and scaled to obtain values within the range -1, 1.~~

-  For each grid point $(x_i, y_j)$ enclosing $(x, y)$, a distance vector $\mathbf{d}_{i,j} = (x - x_i, y - y_j)$ is generated.

- Finally, the noise function is the sum of dot products, $N(x,y) = \sum_q w_q (\mathbf{g}_{ij}^q \cdot \mathbf{d}_{i,j}^q)$, for $q$ grid points surrounding $(x, y)$. Weights $w_q$ correspond to $w_q = C \frac{1}{\|\mathbf{d}_{i,j}^q\|}$, and $C$ a normalization constant to ensure that $N(x,y) \in [-1, 1]$.

~~When clouds, rain or atmospheric conditions affecting the concentration of aerosols in the air enter the measuring domain, what we see are regions with anomalous line-of-sight wind speeds and low CNR values. One of the advantages of procedural noise is the generation of areas with noise, with the same size of the scan domain for instance, either in polar or Cartesian coordinates, via an array of points $p$ distributed over the synthetic measuring domain. In this test, the distribution of points $p$ over the scanning area aims to follow the decay in theintensity with distance(See Figure 4). We define three bands per scan,,and with a width of% of the beam length in the radial direction. Within each of these bands a set of uniformly distributed positions are selected. The fraction of beams contaminated at each band, as we depart from the lidar, are 30, respectively, since it is assumed that after a position of a particular beam is sampled, the remaining points in the radial direction are all contaminated. The increasing fraction of contaminated points tries to be similar to what is observed in real data, since the quality of the signal decrease as we depart form the lidarfinally scaled by; i.e. ,of $V_{LOS}$~~for the instruments described in Section 2.

Figure 2 ( c) show one contaminated scan and its increasing contaminated area as we move along the beams. The same Figure shows the distribution of the noise generated by the  algorithm after scaling, and  the probability distribution of contaminated synthetic $V_{LOS}$ compared to real data with low values of CNR. The distribution of  real data presents heavier tails than the ones generated, with higher probability of observing extreme values of $V_{LOS}$. Modeling real noise is difficult,

[Figure]

**Figure 2.**  Procedural noise on synthetic scans.(a) Distribution of $\Delta V_{LOS,noise}$, the noise added . Maximum values are within the observable range between  [-35, 35] [m/s]. (b) Distribution of  real $V_{LOS}$ with low CNR values (black) and contaminated, synthetic $V_{LOS} + \Delta V_{LOS,noise}$ (red)    a mean wind direction facing the scan. (c) Individual scan showing  the  increasing fraction of added noise $\Delta V_{LOS,noise}$(grey) with distance.

to spot

since the process that generates it depends on the measuring principle of
the lidar and atmospheric conditions. The synthetic noise used here does not intend to be totally realistic, but more subtle and
smoother than the one observed in real measurements, making the identification of contaminated points more difficult.

**4 Filtering techniques  applied on real and synthetic data**

**4.1 CNR threshold**

CNR thresholds are well known and lidar manufacturers usually
recommend values for rejection of signals with poor backscattering or hitting hard targets (Cariou, 2015).

measurements will lay in between these too scenarios, and thresholds values have traditionally been used to filter out non

250 reliable observations (Gryning et al., 2016). Observations with CNR values lying beyond upper and lower thresholds (which will depend on the lidar itself, commonly -8 and -24 dB, respectively) are rejected. One problem that arise with the use of this filtering criteria for long range lidars is that the CNR value worsen rapidly with distance, and observations which might be valid at spatial points that are distant from the instrument are rejected using this criteria. Figure 3 (b) shows a scatter plot of radial wind speed, $V_{LOS}$, against CNR, from 30 consecutive PPI scans from the Balconies experiment, with its typical comb-like

255 shape. Below the lower threshold (in this case However, the selection of an appropriate threshold for CNR that assures data quality and good data recovery is not easy. Figures 3 and 4 show data from a scan with noisy observations from CNR values below -27 dB) along with extreme . Both, extreme and limited values of $V_{LOS}$we also find line-of-sight speeds that are not far from the reliable range above the threshold. The effect of filtering out only observations below the threshold or accepting the the ones with values within the range of $V_{LOS}$ above the threshold can be seen in Figure 3 (c)and (d). Low CNR values for

260 reasonable line-of-sight speeds can be understood when the range distance is included in the picture, as in , show low CNR values in the distant region of the scan, and data loss results after the application of the CNR threshold (Figure 4 (b)). When a limit to $V_{LOS}$ is applied instead, Figure 4 : the decay in the back-scattering intensity with distance makes CNR values worse, but, apparently, the lidar is still able to retrieve some valid observations. The selection of an appropriate threshold for CNR is not clear, but a lower bound of -29 dB is recommended by Gryning and Floors (2019) before a wind resource assessment,

265 based on lidar measurements, starts to be dependent on the CNR-threshold(c) show that the smoothness in $V_{LOS}$ is lost in the lower part of the scan. A conservative threshold of -24 dB is used here, since the resulting $V_{LOS}$ probability distribution show very little outliers and it can be used as a reference when the performance of the filters proposed are compared.

[Figure]

**Figure 4.** (a)  Un-filtered scan with $V_{LOS}$ values outside the range [-21, 0] in Figure 3 (a) in black.  (b) Filtered scan with  CNR  > -27 and the  resulting data loss in the uper part of the scan. (c)  $V_{LOS}$ within the [-21,0] range,  showing anomalous values in the lower part of the scan.

[Figure]

**Figure 3.** (a)  CNR and $V_{LOS}$ for  one scan from Balconies experiment, including the probability density  (KDE). Observations with  CNR >-27 dB (dashed red line) show a limited range of $V_{LOS}$ ( dashed black line). A  portion of  observations  with high probability density  remain in the rejection area. (b)  CNR v/s distance for the same data. Observations with low CNR  values and high probability density can be found in the  distant region of the scan

**4.2 Median-like filter**

The  median filter arise as a viable option for detecting erroneous measurements, since it is well known that this type of non-linear filter is suited to detect and filter noise that present distributions with large tails . Here we use an adaptation of the traditional median filter used in the  image-processing community, closely related to the three-stage filtering technique described in Menke et al. (2019): observations are not replaced by the local moving median but  excluded if the absolute difference between  their value and the  local moving median is above a certain threshold, $\Delta V_{LOS,threshold}$. Unlike Huang et al. (1979), The two-dimensional moving window is replaced by a two one-dimensional  instances, the first in  line-of-sight or radial direction, $r$, and finally in the azimuth direction, $\theta$, considering the polar coordinates of the scan. This simplification reduces the computation time  importantly.

The input parameters of this filter will be the size (or number of elements) of the moving  windows in the radial  and azimuth directions, $n_r$  and  $n_\phi$ respectively, and $\Delta V_{LOS,threshold}$. For fixed values of $\Delta V_{LOS,threshold}$, $n_r$ and $n_\phi$, the spatial structure of wind speed fluctuations will have an important effect on the recovery rate and noise detection of this filter.

**4.3**

~~Assuming no abrupt spatial changes in the different features measured, radial wind speed for instance, observations non affected by poor back-scattering or noise generated in the lidar itself, will fall in limited regions of the observational space, unlike contaminated observations, that will be scattered in wider region. This is noticeable in Figure 4, distinguishing two main groups: one presenting limited radial wind speeds and CNR values (between -20 and 0and 0 and -35 dB, respectively) and another group, less dense and broader, with bad observations. Also noticeable is an overlapping region that contains observations from both groups. When only $V_{LOS}$ and CNR are taken into account (Figure 3 (b)), the overlapping region is more diffuse than when including the range gate distance in the picture (Figure 4). The two groups are more distinct in the latter representation because their separation increases along with dimensionality as we consider more features of the data. These considerations inspire us to think the identification of valid observations as a process, which classifies groups or *clusters* of good/bad data, using all the information available: temporal~~

300  (See Appendix A). This set is used both for artificial and real data. The filter does not include a time window, and it is applied on individual scans.

305 ~~these algorithms data points are separated in $k$ different groups or clusters, with each observation belonging to the cluster with the closest mean/medoid, estimated iteratively by minimization of the within-cluster variance. Even though these algorithms introduce the notion of density in the data distribution, they have two drawbacks: both need prior knowledge of the number of clusters present in the data and, if this first problem is sorted, the algorithm assigns all data points to specific clusters, either good measurements or outliers, which is not desirable for our purpose. The Density Based Spatial Clustering for Applications~~
310 ~~with Noise algorithm, or `DBSCAN` (Ester et al., 1996), on the other hand, is a clustering technique specially designed to deal with large-scale data with spatial distribution. When compared to other algorithms, `DBSCAN` presents several advantages when applied as a filter to measurements from lidars: 1) it can manage large amounts of data with spatial distribution (its time complexity is $O(n \log n)$, with $n$ the amount of~~

**4.3 Filtering using a clustering algorithm**

315 If we represent lidar observations as $m$-dimensional  vectors, with $m$ the number of features/parameters of the data, measurements not affected by poor back-scattering or noise will cluster together in regions of high data density, as shown in Figures 4 and 3. The approach
320 presented here identifies such clusters applying `DBSCAN` on data described by CNR, $V_{LOS}$ and, additionally, spatial location and smoothness features, which help to make clusters more distinguishable.

`DBSCAN` identifies clusters and noise based on two parameters,  : a neighbourhood size, $\varepsilon$, and  a minimum number of nearest neighbours,  $NN$. The parameter $\varepsilon$ is the euclidean distance from one observation to the limits of a neighborhood  that might contain $NN$ (or more) nearest neighbors . Intuitively, these parameters
325 will define the minimum density that a  group of data points needs to have to be identified as a cluster.  Observations within a cluster fall into the following categories,

- Core point:  points $q$  whose $\varepsilon$-neighborhood  contains $NN$ or more points .

330 - Direct density reachable point:  points $p$  which are reachable by $q$  by laying within its $\varepsilon$-neighborhood .

- Density reachable point:  points $p$ is  reachable by a point $r$  through one or a set of  to $r$ and $p$.~~

-  directly connected core points $q$.

 DBSCAN

 DBSCAN  DBSCAN ~~algorithm working: (b) The current point being evaluated have the minimum number of nearest neighbours required, NN, within a neighborhood of size $\varepsilon$, classified as a *core point* (red) (c) The next point have less than NN neighbours, but one of them is a core point and becomes a *border point* (yellow) (d) A point with neither NN neighbours, nor core points within $\varepsilon$, classified as *noise* (brown) (e) The final cluster and noise. The former is a collection of density connected points.~~

 DBSCAN  travels across data points identifying  core points,  border points (density reachable points  with at least one core point within the $\varepsilon$-neighbourhood ) and noise, or points that  do not belong to any of the categories described above. Finally, the algorithm  separates clusters as individual groups of  density connected points . Figure 5 shows schematically these definitions and how the algorithm works.

The input parameters $\varepsilon$ and  $NN$ have a significant influence on the number and characteristics of the clusters detected  . For example, large $\varepsilon$ together with   a small $NN$ will end up with sparse clusters that might include noise.  In order to find the parameters separating the least dense cluster from noise, we fix $NN$ to a certain value $k$ and determine $\varepsilon$ from the data density distribution. The latter is well described by the *k-distance* function, $d_k(n)$ , which represents the distances from all data point $n$ to their

[Figure]

**Figure 5.** (a) `DBSCAN` algorithm definitions: direct density reachable point $p$ (reachable by the core point $q$) and density reachable and density connected points $p$ and $r$. Here point $n$ does not belong to any of these categories, but noise. The `DBSCAN` algorithm working: (b) The current point being evaluated have the minimum number of nearest neighbours required, NN, within a neighborhood of size $\varepsilon$, classified as a *core point* (red) (c) The next point have less than NN neighbours, but one of them is a core point and becomes a *border point* (yellow) (d) A point with neither NN neighbours, nor core points within $\varepsilon$, classified as *noise* (brown) (e) The final cluster and noise. The former is a collection of density connected points.

respective $k$-th nearest neighbour. , sorted in ascending order. When $k = NN =$  5  for instance, $d_5(n)$ in Figure 6 shows sudden changes (or knees) that give some indications about the data density distribution. The knee highlighted represents a limit between a group of reliable observations and  the one growing fast towards noisy data. The positions of these knees in the graph correspond to the peaks in the curvature of the $d_k(n)$, $\kappa(n)$  expression (3) . In this expression primes correspond to the derivatives of $d_k(n)$ with respect to  $n$. The continuous  version of $d_k(n)$ is made by spline-fitting on a reduced set of uniformly distributed points over the original data set.

$$\kappa = (d_k(n))'' / (1 + (d_k(n))'^2)^{3/2} \tag{3}$$

[Figure]

**Figure 6.** (a) Scan from the Balconies experiment (phase 1) with a 48% of data points in the range of reliable observations with CNR $\in$ [-24, -8] dB (b)  same scan after filtering total number of observations corresponds to three consecutive scans, or 26730 points. The sorted 5-*th* distances show three knees separating three types of structures: reliable observations with  $V_{LOS}$ distances below $\varepsilon_{knee}$,  an overlapping region where the distance between points grows faster,  and  pure noise or non structured data.

380       When scans are very noisy, the selection of a proper value of $\varepsilon$ is difficult, since knees are located closer together and a larger fraction of observations show a fast growing  is difficult but can be eased when $d_k(n)$, as expected. In this case,

385 the fraction of points  showing a reliable CNR  values is taken into account  and $\varepsilon$ is estimated by expression (4). Here $f_{CNR}$ corresponds to the fraction of  observations CNR values within the range [-24, -8]  and the constants $c_1$ and $c_2$ are  obtained obtained from the upper and lower bounds of $\varepsilon$

390  in the data, respectively.

$$\varepsilon_{CNR} = c_1 f_{CNR} + c_2 \qquad (4)$$

Data structure of the scan shown in Figure ??. (a) Logarithm of sorted distances to the 5-*th* nearest neighbour for each point in a data set. The total number of observations corresponds to three consecutive scans, or 26730 points. The sorted 5-*th* distances show three knees separating three types of structures: reliable observations with distances below $\varepsilon_{knee}$, an overlapping region where the distance between points grows faster and pure noise or non structured data. (b) Tri-dimensional representation of the data (range gate, CNR and $V_{LOS}$), where two coherent structures of data points, or clusters, are identified. The large structure has distances between observations below $\varepsilon_{knee}$ and corresponds to reliable observations. The vertical line showing high CNR values are observations of bad $V_{LOS}$ measurements in a group of 7 turbines around 2000 m from the lidar (see Figure 13).

(a) Scan from phase 1 of the experiment with a 13% of data points in the range of reliable observations with CNR $\in$ -24, -8dB (b) Data structure of the very noisy data. Here $\varepsilon_{knee}$ over estimates the neighbourhood distance for a coherent cluster, and the inclusion of noisy measurements is avoided via $\varepsilon_{CNR}$

Regarding the features considered to characterize each data point, depending on whether we filter synthetic or real data , these will be radial or line-of-sight wind speed, $V_{LOS}$, The set of features considered when filtering synthetic data does not include CNR, because it is not available from the the lidar simulator described in Section 3. For synthetic and real data sets we consider spatial location (azimuth and radial positions and $\Delta V_{LOS}$) and smoothness as additional features. The latter, $\Delta V_{LOS}$, corresponds to the median difference in radial wind speed of an specific radial and azimuth position with $V_{LOS}$ between a specific position and its direct neighbours . This feature is included to consider the smooth spatial fluctuations of the radial wind speed, same assumption used by the median-like filter . The real data will include $CNR$ as an additional feature. The controlled performance test on synthetic data does not include this feature, because the numerical lidar described in Section 3 just estimates $V_{LOS}$. in one individual scan.

Since we consider features that vary importantly in magnitude (CNR and range gate distance for instance), we normalize the data before the application `DBSCAN`. This step is necessary for the estimation of meaningful distances between observations, basis of this approach. There are several ways to do this. Here, the data in each feature is centered by subtracting its median, and scaled according to its inter-quantile range. This aims to minimize the influence from outliers in the normalization.

The clustering filter is implemented to be a non-supervised classifier, and does not need more input parameters than the different features and the number of scans put together as a batch before filtering. The latter is set to three in this case, to speed up calculations and avoid creating clusters from noisy regions. From this point of view, this filter is also dynamic as that of Beck and Kühn (2017) when applied to a real data set, since it will consider the data structure within a period limited to 135 seconds (3 scans of 45 seconds in our case), and characteristics of temporal evolution of the data is indirectly taken into account. For the synthetic data used in this test, more than one scan filtered per iteration gives enough data density in noisy and reliable areas of the observational space. We speculate that scans that are correlated in time will enhance the self-similarity of the data, thus improving the performance of the filter. Turbulence structures with length scales in a range between the range gate size and the scanning area size will evolve at a slower rate than the time elapsed between consecutive scans.

**5 Performance metrics**

**5.1 Synthetic data**

Expressions (5) to (7) defines three metrics to assess the performance of the filters, given prior knowledge of the position and magnitude of noise in a controlled case with $N$ observations. The fraction of noise detected, $\eta_{noise}$, quantify the relative importance of true positives, or the difference between observations identified as noise, $N_{noise}$, and false positives, $N_{pos}$, over the total number of contaminated observations. The fraction of good observations recovered, $\eta_{recov}$, give an idea of the true negatives over the total number of non-contaminated observations, $N_{non-cont}$. True negatives are not equal to $N - N_{noise}$, since the latter might include false negatives, $N_{neg}$. The relative importance of this two metrics, for a given fraction of noise in a contaminated scan, $\eta_{noise}$, is quantified by $\eta_{tot}$, which takes into account cases with a large fraction of noise detected and low recovery rate, and $\eta_{recov}$ vice-versa.

$$\eta_{noise} = \frac{N_{noise} - N_{pos}}{N_{cont}} \tag{5}$$

$$\eta_{recov} = \frac{N - (N_{noise} + N_{neg})}{N_{non-cont}} \tag{6}$$

$$\eta_{tot} = f_{noise}\eta_{noise} + (1 - f_{noise})\eta_{recov} \tag{7}$$

**5.2 Real data**

[revised manuscript text omitted]

the median-like filter. The PDF of $V_{LOS}$ in this area also show more similarities between the data filtered with the clustering
algorithm and observations with CNR values in [-24,8].

Table 5 shows a summary of the additional data available when the CNR = -29 dB threshold, the median-like and the clustering filters are applied instead of the more conservative and restrictive CNR = -24 dB threshold filter. Additionally, this table shows the fraction of observations exceeding the 3-$\sigma$ limit that are recovered by the three filters. Even though the clustering filter shows a slightly lower fraction of additional data available when compared to the other filters, most of it comes from values within the 3-$\sigma$ region. Moreover the quality of the data recovered by the clustering approach seems to be higher when all these results are tested with the performance metrics defined in Section 5.

**7 Discussion**

**7.1**

The metrics introduced in section  5.1 attempt to evaluate two different capabilities of the filters: the quality   and amount od the data recovered. In general these two metrics are in

[revised manuscript text omitted]